# Laser ablation of Dbx1 neurons in the pre-Bötzinger complex stops inspiratory rhythm and impairs output in neonatal mice

Xueying Wang[1†‡], John A Hayes[1†], Ann L Revill[2,3], Hanbing Song[1], Andrew Kottick[1], Nikolas C Vann[1], M Drew LaMar[4], Maria Cristina D Picardo[1], Victoria T Akins[1], Gregory D Funk[2,3], Christopher A Del Negro[1*]

[1]Department of Applied Science, The College of William and Mary, Williamsburg, United States; [2]Department of Physiology, University of Alberta, Edmonton, Canada; [3]The Women and Children's Health Research Institute, University of Alberta, Edmonton, Canada; [4]Department of Biology, The College of William and Mary, Williamsburg, United States

*For correspondence: cadeln@wm.edu

†These authors contributed equally to this work

Present address: ‡Department of Neurology, Massachusetts General Hospital, Charlestown, United States

**Abstract** To understand the neural origins of rhythmic behavior one must characterize the central pattern generator circuit and quantify the population size needed to sustain functionality. Breathing-related interneurons of the brainstem pre-Bötzinger complex (preBötC) that putatively comprise the core respiratory rhythm generator in mammals are derived from *Dbx1*-expressing precursors. Here, we show that selective photonic destruction of Dbx1 preBötC neurons in neonatal mouse slices impairs respiratory rhythm but surprisingly also the magnitude of motor output; respiratory hypoglossal nerve discharge decreased and its frequency steadily diminished until rhythm stopped irreversibly after 85±20 (mean ± SEM) cellular ablations, which corresponds to ~15% of the estimated population. These results demonstrate that a single canonical interneuron class generates respiratory rhythm and contributes in a premotor capacity, whereas these functions are normally attributed to discrete populations. We also establish quantitative cellular parameters that govern network viability, which may have ramifications for respiratory pathology in disease states.

## Introduction

Central pattern generator (CPG) circuits give rise to common behaviors such as swimming, walking, and breathing (*Grillner, 2006*; *Grillner and Jessell, 2009*; *Kiehn, 2011*). To understand the cellular origins of these behaviors, key problems are to identify the rhythmogenic and premotor populations, and then quantify the requisite number of neurons to sustain functionality. We address these issues in the mammalian breathing CPG by cumulatively ablating a genetically identified interneuron population hypothesized to form the rhythmogenic core while monitoring effects on network output in real time.

The brainstem pre-Bötzinger complex (preBötC) putatively drives inspiratory breathing rhythms (*Smith et al., 1991*; *Feldman et al., 2013*; *Moore et al., 2013*). These rhythms persist in reduced slice preparations that retain the preBötC and can be monitored via respiratory hypoglossal (XII) nerve discharge, providing a powerful in vitro model of breathing behavior (*Lieske et al., 2000*; *Koizumi et al., 2008*; *Funk and Greer, 2013*). Rhythmogenic preBötC interneurons are distinguished by glutamatergic transmitter phenotype or the expression of neuropeptides and peptide receptors, or their intersection (*Funk et al., 1993*; *Gray et al., 1999*, *2001*; *Guyenet et al., 2002*; *Stornetta et al., 2003*;

**eLife digest** Our first breath, moments after we are born, is the result of a pattern of activity in our brain that started in the embryo and will continue almost effortlessly until we die. Like other rhythmic activities, such as walking and swimming, breathing originates from circuits of neurons in the brain that generate patterns. These circuits pass messages to other cells that translate them into the physical movements required to take a breath. Interrupting these patterns by injury or illness can lead to breathing disorders or cause death.

Previous studies have identified a class of neuron, which all express a specific gene, that is necessary for breathing. Mice born without this class of cell failed to ever take a breath and died at birth. These neurons are found in part of the brainstem and can continue to generate rhythm even when this section of the brainstem is removed from newborn mice and cut into very thin slices. However, it is unclear how many of these neurons are needed to maintain a breathing rhythm.

Wang et al. used a laser to destroy the breathing rhythm-generating neurons in these slices one at a time and found that the rhythm of breathing in (i.e., inspiration) stopped after ~15% of the neurons were destroyed. This suggests that a high percentage of these neurons must be maintained for breathing to continue normally.

Wang et al. also discovered that destroying the rhythm-generating neurons reduced the strength of the signals sent from the brainstem to trigger the movements that cause breathing in. This suggests that the same class of neurons also sends messages to the muscles involved in breathing; it was previously thought that a separate class of cell in the same part of the brain sent these messages.

Studies involving live animals are now needed to confirm the results. If confirmed, the findings may be used to develop new treatments for a number of breathing disorders. Medications that boost the signals sent to the muscles by these neurons might be useful for treating sleep apnea. Wang et al. also suggest that medications that boost rhythm generation might be useful for premature infants with breathing difficulties and people with drug-induced breathing problems. Moreover, finding ways to maintain breathing rhythms with fewer of these neurons may help those with neurodegenerative disorders, which cause cells in the brain to be lost.

*Wallén-Mackenzie et al., 2006*; *Tan et al., 2008*). Glutamatergic, peptidergic, and peptide receptor-expressing preBötC interneurons develop from a common set of precursors that express the home-obox gene *Dbx1* during embryonic development. *Dbx1*-derived interneurons (hereafter Dbx1 neurons) in perinatal mice are inspiratory modulated, and *Dbx1*-null mice die at birth without ever breathing (*Bouvier et al., 2010*; *Gray et al., 2010*; *Picardo et al., 2013*). Therefore, we—and others—proposed that Dbx1 preBötC neurons comprise the core inspiratory rhythm generator, i.e., the Dbx1 core hypothesis.

Previously, we estimated the cellular mass critical for respiratory rhythm generation by laser-ablating preBötC inspiratory interneurons identified by $Ca^{2+}$ imaging. The destruction of ~120 rhythmic neurons irreversibly stopped respiratory rhythmogenesis (*Hayes et al., 2012*). However, inspiratory-modulated preBötC neurons may be excitatory or inhibitory. $Ca^{2+}$ fluorescence changes cannot differentiate rhythmogenic glutamatergic neurons (*Funk et al., 1993*; *Wallén-Mackenzie et al., 2006*) from GABA- or glycinergic neurons (*Kuwana et al., 2006*; *Winter et al., 2009*), which influence sensory integration and coordinated patterns of muscle contraction for respiratory behaviors, but are non-rhythmogenic because inspiratory rhythms in vivo and in vitro do not require synaptic inhibition (*Feldman and Smith, 1989*; *Brockhaus and Ballanyi, 1998*; *Kuwana et al., 2003*; *Ren and Greer, 2006*; *Fujii et al., 2007*; *Janczewski et al., 2013*). Therefore, we predicted that the selective destruction of Dbx1 preBötC neurons, which are predominantly glutamatergic (*Bouvier et al., 2010*; *Gray et al., 2010*)—unlike locomotor Dbx1 neurons in lumbar spinal cord of which ~70% express inhibitory transmitters (*Lanuza et al., 2004*; *Talpalar et al., 2013*)—would impair rhythmogenesis with a cell-ablation tally much lower than 120 (*Hayes et al., 2012*). To test this prediction of the Dbx1 core hypothesis, we used photonics to map the positions of Dbx1 preBötC neurons, and then laser ablated them—one cell at a time—while continuously monitoring respiratory motor output (*Wang et al., 2013*). As predicted, cumulative destruction of Dbx1 preBötC neurons progressively decreased respiratory

frequency until rhythm ceased after ablation of ~85 neurons. Surprisingly, cumulative Dbx1 cellular ablations also diminished the amplitude of respiratory XII nerve discharge, suggesting that Dbx1 preBötC neurons also influence motor output. In simulations that assign only rhythm-generating function to preBötC neurons, cumulative ablations decreased frequency and stopped rhythmogenesis but at much lower tallies and without perturbing the output amplitude. These ablation results and model-experiment discrepancies, combined with antidromic activation from the XII nucleus and axonal projection patterns, ascribe rhythm-generating and premotor roles to Dbx1 preBötC neurons. Thus, we demonstrate that one cardinal class of hindbrain interneurons serves two distinct roles in a key mammalian CPG; the data further establish quantitative cellular parameters that minimally ensure network functionality.

## Results

### Ablation of Dbx1 preBötC neurons precludes rhythmogenesis

Dbx1 neurons were detected and mapped within the preBötC, and then laser ablated individually, in sequence, while monitoring respiratory network functionality via XII motor nerve output. Experiments began with an 'initialization phase' that defined the domain for detection and ablation, which was bilateral. We used a *Dbx1* Cre-driver line (*Dbx1^CreERT2^*) coupled with floxed reporter mice (*Gt(ROSA)26Sor^flox-stop-tdTomato^*) to locate Dbx1 neurons via fluorescence. Viewed in the transverse plane of slices that expose the preBötC at their surface (i.e., preBötC-surface slices), Dbx1 neurons form a bilaterally symmetrical V-shape starting dorsally at the border of the XII motor nuclei and continuing ventrolaterally to the preBötC (*Figure 1A* and *Figure 1—figure supplement 1A*). Dorsally, the preBötC adjoins the semi-compact division of the nucleus ambiguus (scNA); the ventral border of the preBötC is orthogonal to the dorsal boundary of the principal sub-nucleus of the inferior olive (IOP$_{loop}$) (*Ruangkittisakul et al., 2011*, *2014*). These spatial relationships visible in bright field or epifluorescence allow us to pinpoint the preBötC (*Figure 1A*). At the cellular level, identifying putative rhythmogenic neurons on the basis of fluorescent protein expression alone is acceptable because the overwhelming majority of Dbx1 preBötC neurons are inspiratory (e.g., *Figure 1B*) (*Picardo et al., 2013*).

In the subsequent 'detection phase', a visible wavelength laser scanned the domain and an iterative threshold-crossing algorithm analyzed the image to then draw regions of interest (ROIs) for putative cell targets based on fluorescence brightness. Potential targets were evaluated on the basis of shape to differentiate cell bodies from auto-fluorescent debris, and to reject the fluorescence from dendrites and neuropil whose somata were detectable in adjacent focal planes (*Wang et al., 2013*) (*Figure 1—figure supplement 2*). The map of ROIs for validated cell targets was retained at each focal plane (*Figure 1C—figure supplements 1B and 2*, red ROIs). Potential targets that did not meet these criteria were discarded (but displayed for demonstration purposes in *Figure 1C* and *Figure 1—figure supplements 1B and 2*, blue ROIs). Target detection was repeated at 10-µm increments through the z-axis and the final three-dimensional map of targets was stored in memory (*Figure 1D*). Typically, we detected 26–50 Dbx1 neurons per focal plane per side (*Figure 1—figure supplement 3*) for a total average number of 705 targets in the preBötC (SD 119, SEM 59, range: 548 to 802, *n* = 8 slices).

During the 'ablation phase' of the experiments, Dbx1 neurons in preBötC-surface slices were randomly selected for photonic lesioning. Each target in the domain was individually spot scanned with a Ti:sapphire laser using maximum intensity 800-nm pulses until target destruction was confirmed by three forms of optical criteria (*Wang et al., 2013*) or was deemed a failure. Generally >90% of lesion attempts are successful (*Figure 2—figure supplement 1*) (*Hayes et al., 2012*; *Wang et al., 2013*). Only confirmed lesions add to the running tally.

The frequency and amplitude of inspiratory motor output diminished at the onset of the ablation phase (*Figures 2A and 3A*). XII amplitude decreased steeply with the tally of ablated cells, and then stabilized at 44% of its pre-lesion value (SD 4%, SEM 1%, suction electrode recordings are reported in normalized arbitrary units). Frequency, however, continued to decrease (i.e., cycle period increased) throughout the ablation phase. Initially, within the first dozen ablations, the average decrease in respiratory frequency was nearly twofold, and it continued to fall steeply until rhythm cessation (range of frequencies: 0.22–0.007 Hz, *Figure 3B* [inset shows bi-exponential increase in cycle period], *n* = 5 slices). Furthermore, the rhythm destabilized during the ablation phase. We defined regularity score (*RS*) as the ratio of the present cycle period with respect to the mean period over 10 prior cycles (see 'Materials and methods' for *RS* formula). Cycle-to-cycle variations in *RS* indicate irregularity; the system

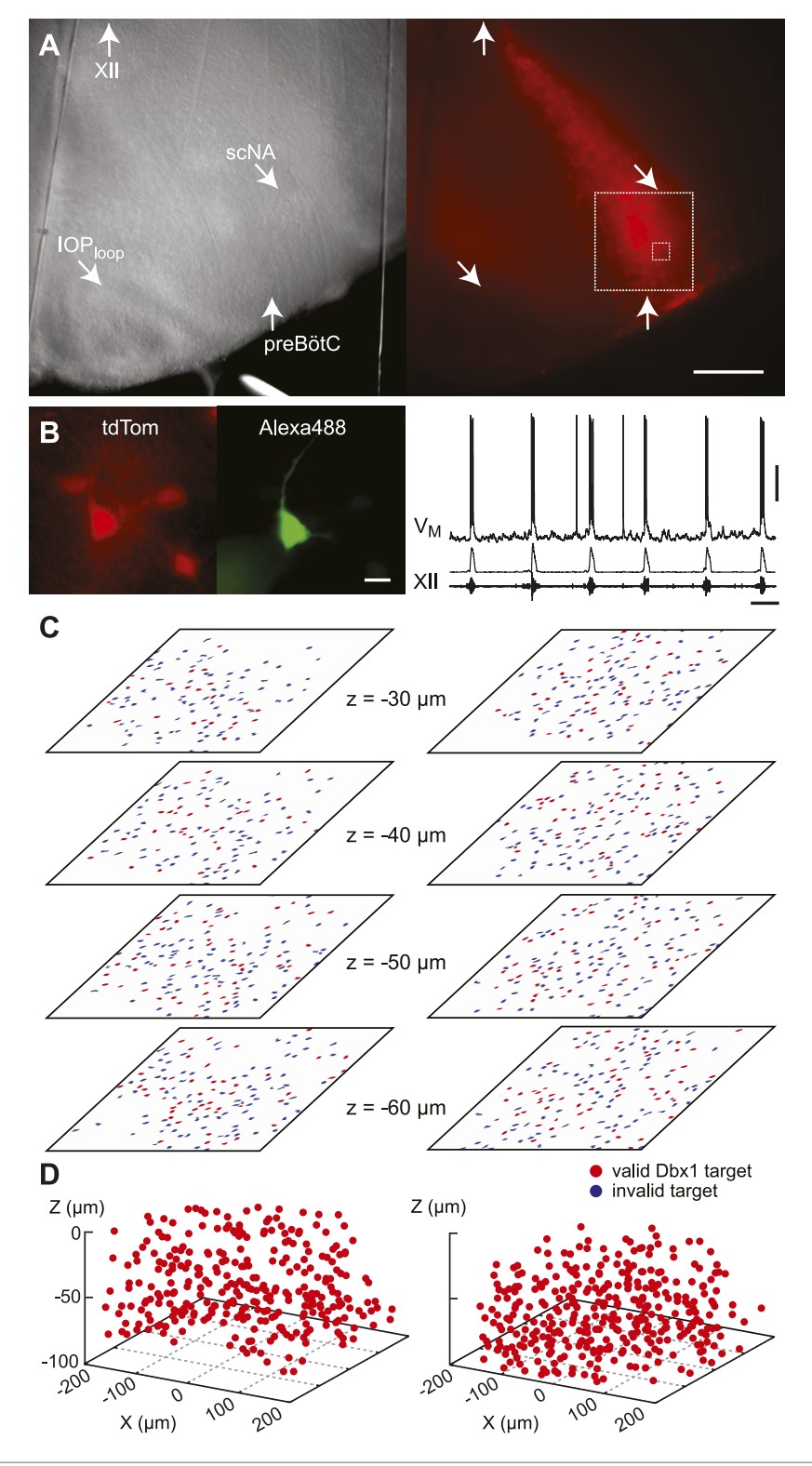

**Figure 1**. Dbx1 preBötC neurons. (**A**) Bright field (left) and fluorescence (right) images of the right half of a preBötC-surface slice preparation. Anatomical landmarks are illustrated including: XII, the hypoglossal motor nucleus; scNA, semi-compact nucleus ambiguus; IOP$_{loop}$, the dorsal loop of the principal inferior olive; and the ventral border of the preBötC, which is orthogonal to the IOP$_{loop}$. Scale bar is 300 μm. At right, the larger white box shows the

*Figure 1. Continued on next page*

*Figure 1. Continued*

detection and ablation domain. (**B**) Expansion of smaller white box in **A**, showing tdTomato expression in Dbx1 neurons and intracellular dialysis via patch pipette with Alexa 488 from the recorded neuron whose robust inspiratory discharge is illustrated at right (scale bar is 10 µm). Respiratory motor output from the XII nerve is shown in raw and RMS-smoothed form. Voltage and time calibration bars represent 20 mV and 2 s. Baseline membrane potential in the recorded neuron was −60 mV. (**C**) Mask of targets showing validated Dbx1 interneuron targets (red) and regions of fluorescence that do not pass muster and were rejected as targets (blue) for focal planes at depths z = (30–60 µm). The region shown in each case maps to the 412 × 412 µm$^2$ square shown by the larger white box in **A** (right). Only a subset of the masks are shown for economy of display. (**D**) 3D reconstruction of detected targets for all focal planes z = (10–80 µm) from the left and right preBötC. Each Dbx1 neuron is represented by a single red point centered on its soma.

The following figure supplements are available for figure 1:

**Figure supplement 1**. Detection of Dbx1 preBötC neurons.

**Figure supplement 2**. Detection of Dbx1 neuron targets via fluorescence and image processing.

**Figure supplement 3**. Average number of Dbx1 neurons detected at each acquisition depth from z = 0 (surface) to z = −80 µm in preBötC-surface slices and control slices with the ventral respiratory column (VRC) exposed at the slice surface.

is trending slower when *RS* exceeds unity. The *RS* of preBötC-surface slices measured 1–9 during the ablation phase (*Figure 3C*). Respiratory rhythm ceased altogether after an average of 85 confirmed Dbx1 neuron ablations in preBötC-surface slices (SD 44, SEM 20, range 42–137, *n* = 5 slices), well before exhausting the average list of 705 targets per slice. These ablations were bilateral and the tally reflects the sum of both sides (*Figure 2—figure supplement 2*). The representative experiment in *Figure 2A* shows rhythm cessation after 62 confirmed ablations, corresponding to 9% of the total 677 detected targets.

## Ablation of Dbx1 neurons from the ventral respiratory column

Detection and ablation were similarly performed bilaterally in control slices whose rostral surface exposed the ventral respiratory column, which occupies a comparable domain for detection and ablation in the transverse plane, but this domain is ~100 µm rostral to preBötC. The ventral respiratory column contains inspiratory and expiratory-modulated neurons that are not associated with rhythmogenesis (*Smith et al., 1990*; *Feldman et al., 2013*).

During the initialization phase in control slices, the domain was centered on the highest density of fluorescent Dbx1 neurons bounded by the compact division of the nucleus ambiguus (cNA) dorsally and the ventral margin of the slice (*Figure 4*). We acquired 38–60 Dbx1 neuron targets per focal plane per side (*Figure 1—figure supplement 3*) for a total average of 906 targets per slice (SD 97, SEM 34, range: 722–1004, *n* = 8 slices).

Dbx1 ventral respiratory column neurons were lesioned in random sequence during the ablation phase of control experiments. The amplitude of XII motor output decreased to 77% of control (SD 2%, SEM 1%, normalized arbitrary units) over the course of the ablation phase (*Figures 2B and 3A*). Frequency did not change. It measured 0.36 Hz (SD 0.2 Hz, SEM 0.04 Hz) during the detection phase compared to 0.39 Hz (SD 0.3 Hz, SEM 0.01 Hz) during the ablation phase, which was not significant (p=0.49, Mann–Whitney *U*-test, *Figure 3B*). *RS* did not deviate from ~1 throughout the ablation phase (*Figure 3D*, *n* = 8 slices). These data indicate that cumulative sequential ablation of Dbx1 ventral respiratory column neurons has no effect on the stability or the period of the respiratory cycle. The ablation protocol exhausted the entire set of targets in every control experiment without stopping the rhythm (e.g., 923 confirmed ablations in *Figure 2B*). Ablations in control slices were also performed bilaterally, where the tally reflects the sum of both sides (*Figure 2—figure supplement 2*).

## Transient recovery of irregular and unsustainable rhythm

Cumulative deletion of Dbx1 preBötC neurons appeared to degrade respiratory oscillator function. Nonetheless, an alternative explanation could involve the loss of excitatory drive (rather than destruction of CPG core circuitry). Dbx1 preBötC neurons express neurokinin-1 peptide receptors (NK1Rs)

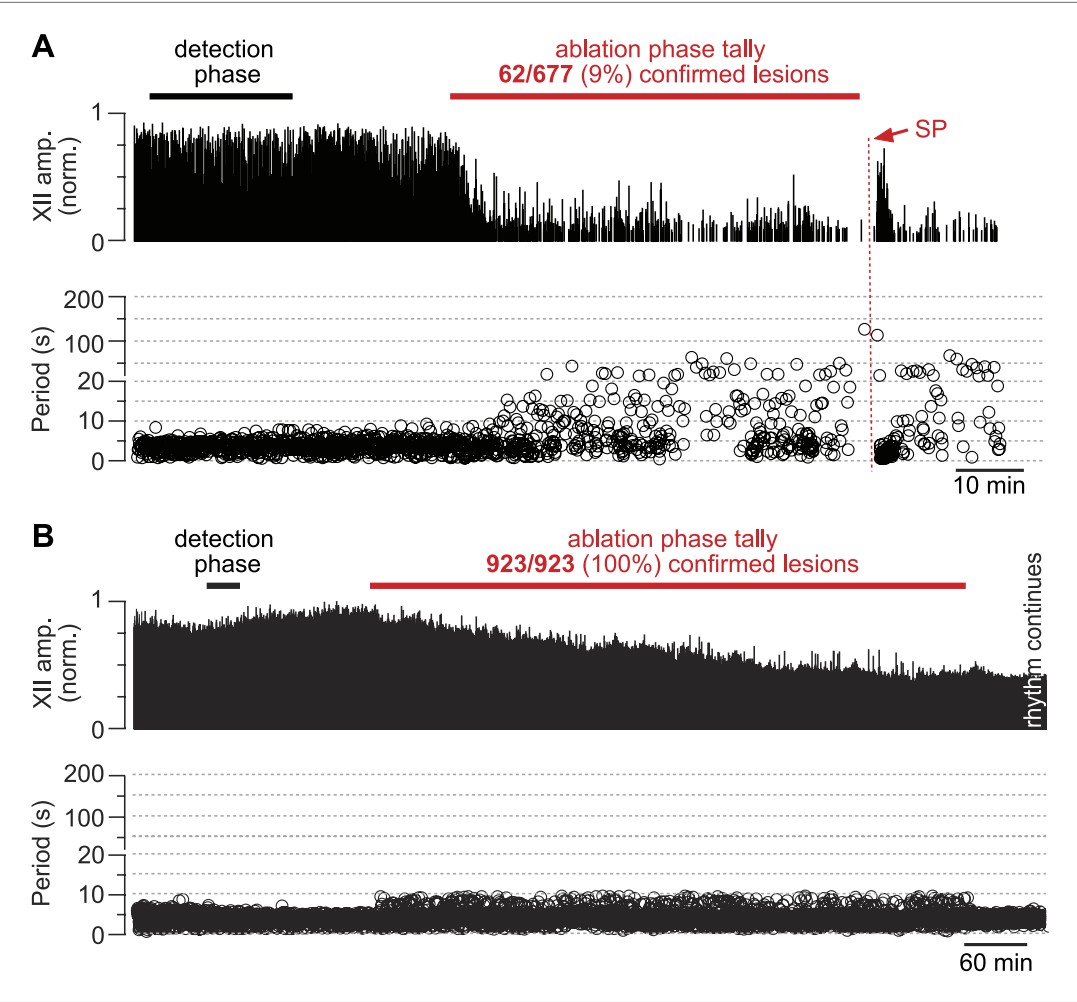

**Figure 2**. Cumulative serial ablation of Dbx1 neurons in preBötC-surface slices (**A**) and control slices whose surface exposes the ventral respiratory column, not preBötC (**B**). (**A** and **B**) The x-axis is a timeline. The y-axis plots XII amplitude (normalized units, top) and respiratory period (bottom). The respiratory period axis is continuous (0–200 s) but plotted with two scales. Major ticks are separated by 10 s from 0 to 20 s (with unlabeled minor ticks at 5 s increments), and thereafter major ticks are plotted in 100 s divisions from 21 to 200 s (with unlabeled minor ticks at 50 s increments). The discontinuity in the y-axis stops at 20 s (lower portion) and starts at 21 s (upper portion). There is one data point for every individual respiratory period measured. The recording in **A** is no longer displayed after 6 min of XII quiescence. Substance P (SP) injection in **A** is displayed at higher sweep speed in *Figure 5C*. The recording in **B** is no longer displayed after 90 min of continuous stable XII output following the end of the ablation phase. Time calibrations in **A** and **B** are shown separately.

The following figure supplements are available for figure 2:

**Figure supplement 1**. Cellular laser ablation and confirmation.

**Figure supplement 2**. Cumulative tally of laser ablations for preBötC-surface slices (magenta) and control slices whose surface exposes the ventral respiratory column, not preBötC (cyan).

---

(*Bouvier et al., 2010*; *Gray et al., 2010*) that stimulate respiratory rhythmogenesis (*Gray et al., 1999*; *Pagliardini et al., 2005*; *Ballanyi and Ruangkittisakul, 2009*). Monoaminergic and peptidergic raphé neurons project to, and receive feedback projections from, the preBötC to elevate excitability in the respiratory network (*Ptak et al., 2009*). Therefore, laser ablation in the preBötC could break the link with the raphé and thus diminish excitatory drive. To test that idea, we exposed the lesioned preBötC to a bolus of neuropeptide substance P (SP, 1 mM) after the respiratory cycle period exceeded 120 s,

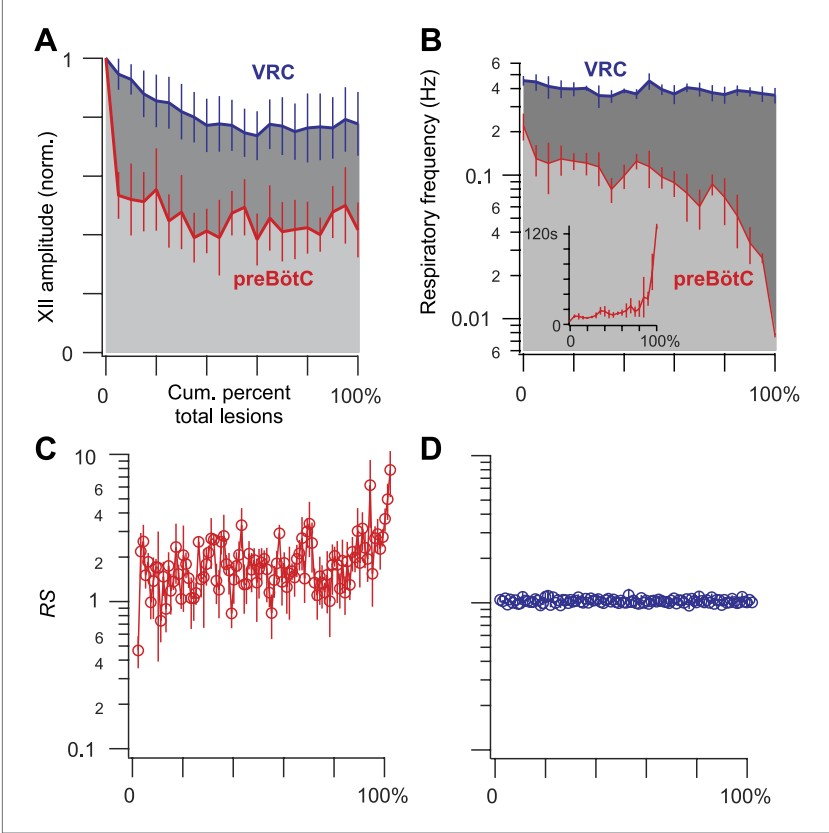

**Figure 3**. Ablation effects on respiratory frequency and the amplitude of XII motor output. (**A**–**D**) Measurements are displayed in light grey and red for preBötC-surface slices and dark grey and blue for control slices that expose the ventral respiratory column (VRC). (**A**) XII amplitude and (**B**) respiratory frequency for preBötC-surface and control slices are plotted vs cumulative percent of total lesions during the ablation phase (bars show SD). Inset in **B** shows respiratory period in lieu of frequency (bars show SD) for preBötC-surface slices. (**C** and **D**) The regularity score (*RS*) is plotted vs cumulative percent of total lesions for preBötC-surface (**C**) and control slices (**D**). **B**, **C**, and **D** are plotted on semi-log axes. **B** and **C** are labeled with subordinate ticks at 2, 4, and 6. Tick labels are omitted from **D** because they match **C** exactly. **B** (inset) has linear axes.

which we previously determined was a reliable benchmark of a slice that would cease rhythmic function within 5–10 min without pharmacological intervention (*Hayes et al., 2012*).

First, as a control, we applied SP to unlesioned preBötC-surface slices, which transiently increased respiratory frequency, i.e., lowered the cycle period from 4.7 s (SD 0.8 s, SEM 0.1 s) to 3.3 s (SD 0.5 s, SEM 0.1 s, average period computed for 25 cycles), and then equilibrated in 21 min (SD 4 min, SEM 2 min, *n* = 4 slices) (*Figure 5A*). The regularity score was ~1 throughout the bout, which is consistent with the stable rhythm expected in unlesioned slices (*Figure 5B*). Then, SP was injected into five preBötC-surface slices wherein the cumulative laser ablation of Dbx1 neurons caused 120 s of quiescence. SP transiently revived respiratory rhythm; the average cycle period was 1.7 s (SD 0.4 s, SEM 0.2 s, computed for 10 cycles after SP bolus injection) but the cycle period slowed down rapidly, surpassing the control period previously measured during the detection phase (4.7 s) within 3 min (SD 2 min, SEM 1 min). Cycle period continuously lengthened and fluctuated from cycle to cycle, and then the rhythm stopped altogether (*Figure 5C*). Judged on the basis of equilibration time, the transient effects of SP were significantly briefer in lesioned slices (p=0.02, Mann–Whitney *U*-test). Furthermore, lesioned slices ultimately fell inexorably silent (*Figures 2A and 5C*), whereas unlesioned slices maintained rhythmicity for 4–6 hr (*Figure 5A*). More importantly, the SP-evoked activity in lesioned preBötC-surface slices was irregular: *RS* measured 2–10 (*Figure 5D*, compare to unlesioned slice in *Figure 5B*). These data indicate that NK1R-expressing Dbx1 neurons can evoke transient cycles of respiratory activity, but the loss of ~85 Dbx1 preBötC neurons slows the respiratory oscillator frequency and renders XII motor output nonfunctional.

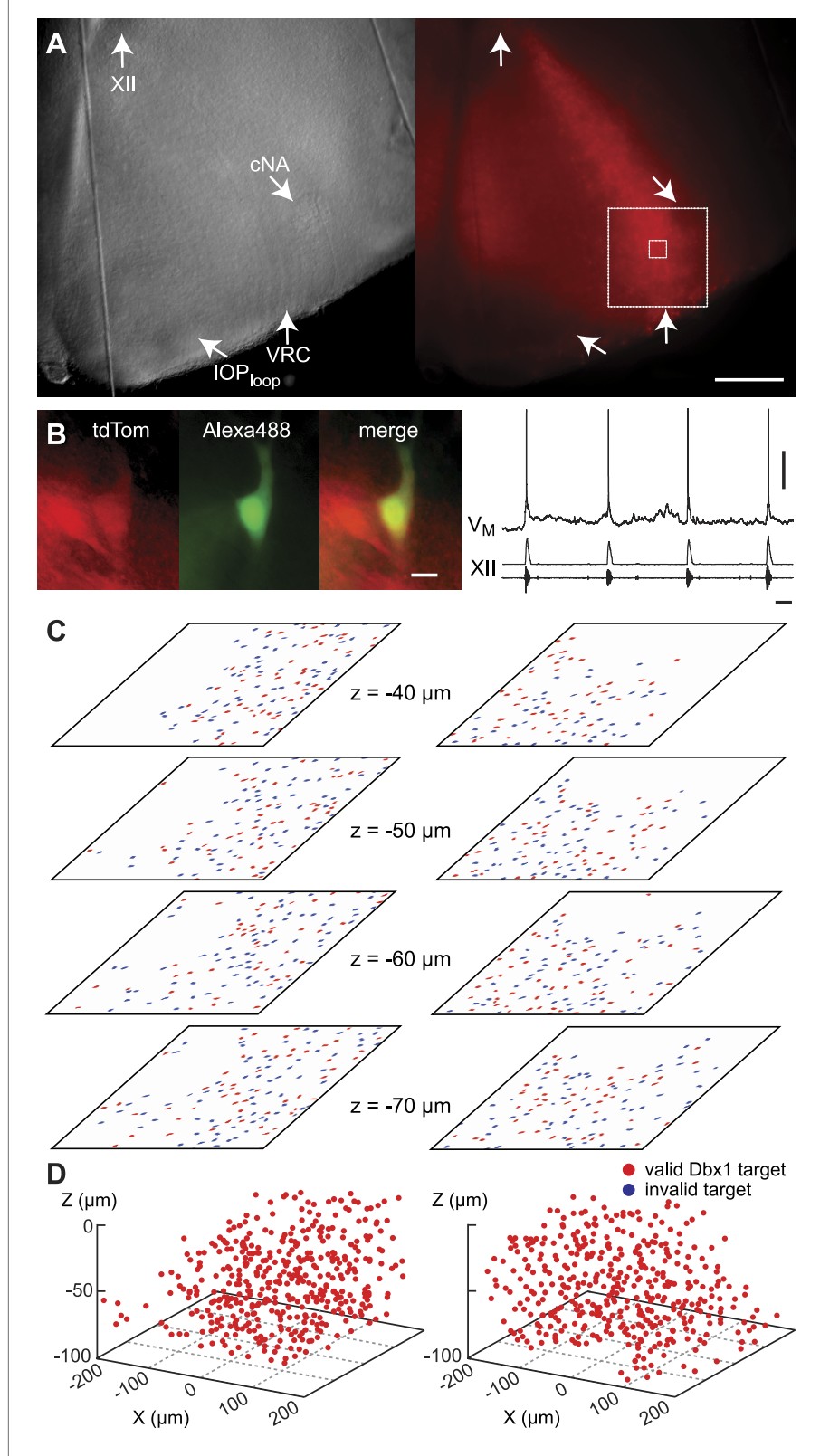

**Figure 4**. Dbx1 neurons in the ventral respiratory column. (**A**) Bright field (left) and fluorescence (right) images of the right half of a control slice preparation. Anatomical landmarks are illustrated including: XII, the hypoglossal motor nucleus; cNA, the compact division of the nucleus ambiguus; IOP$_{loop}$, the ventral portion (loop) of principal
*Figure 4. Continued on next page*

*Figure 4. Continued*

sub-nucleus of the inferior olive; and VRC, the ventral border of the ventral respiratory column. Scale bar is 300 μm. At right, the larger white box shows the detection and ablation domain. (**B**) Expansion of smaller white box in **A**, showing tdTomato expression in Dbx1 ventral respiratory column neurons (scale bar is 10 μm), one of which was recorded. Intracellular dialysis via patch pipette with Alexa 488 is visible in the recorded neuron whose inspiratory depolarization and discharge pattern are illustrated at right. Respiratory motor output from the XII nerve is shown in raw and RMS-smoothed form. Voltage and time calibration bars represent 20 mV and 1 s. (**C**) Masks of targets showing validated Dbx1 interneuron targets (red) and regions of fluorescence that do not pass muster and were rejected as targets (blue) for focal planes at depths z = (40–70 μm). (**D**) 3D reconstruction of detected targets for all focal planes z = (0–80 μm) in the ventral respiratory column (VRC) of the left and right side. A single red point centered on its soma represents each Dbx1 neuron. The highest fraction of accepted Dbx1 target cells is found at deeper focal planes (see *Figure 1—figure supplement 2* and 'priority rule' explained in 'Materials and methods').

## Modeling Dbx1 neuron ablation in the preBötC

We used graph theory and simulations to investigate how Dbx1 neuron ablations affect preBötC structure and function. The Rubin–Hayes preBötC neuron model (*Rubin et al., 2009*) was assembled in Erdős-Rényi G(*n*,p) graphs (*Newman et al., 2006*) with population sizes *n* from 200–400 and connection

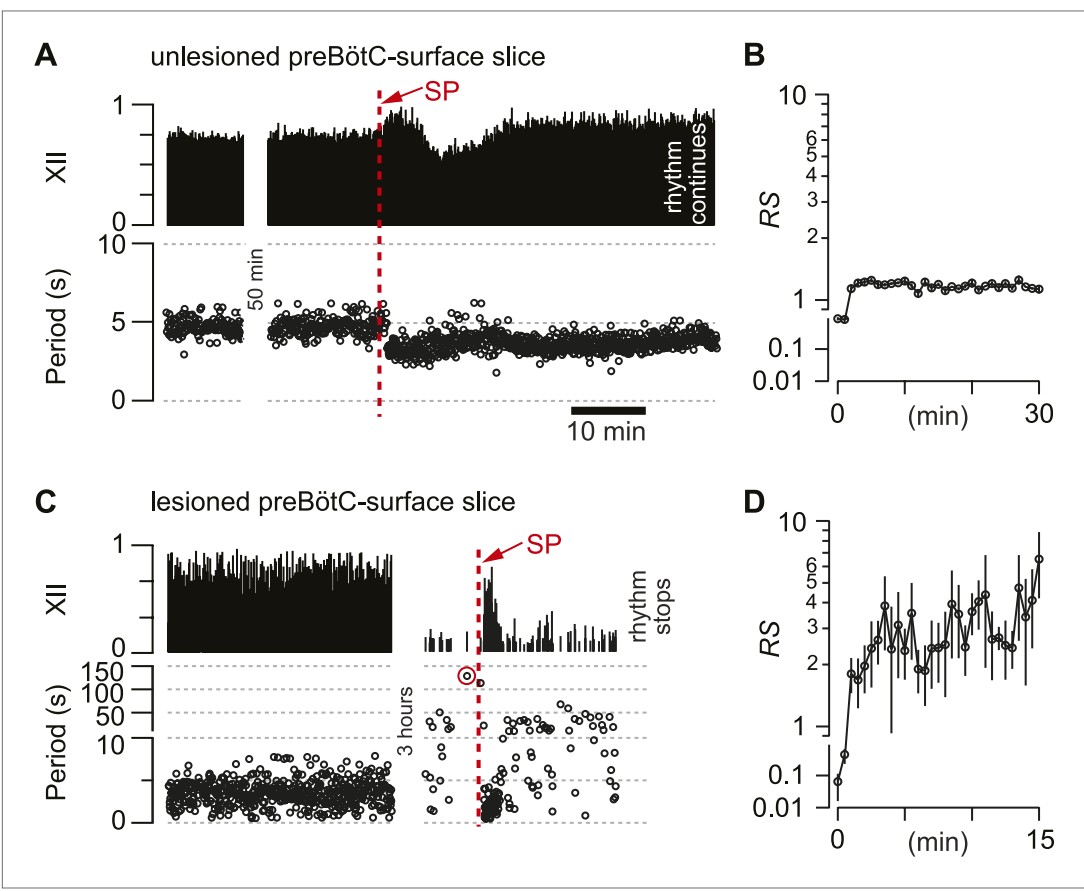

**Figure 5**. Substance-P (SP) injections in preBötC-surface slices. (**A**) SP bolus injected in an un-lesioned preBötC-surface slice. XII output magnitude is plotted with cycle period as a time series. (**B**) Semi-log plot of regularity score (*RS*) for 30 min after SP injection from the slice preparation in **A**. *RS* axis is continuous but plotted with two scales. (**C**) preBötC-surface slice shown in the acquisition phase (left) and during the ablation phase (right), which were separated by a time gap of 3 hr. After 120 s of quiescence (data point circled in red), SP injection revived the rhythm transiently. (**D**) Semi-log plot of *RS* for 15 min after SP injection from the slice preparation in **C**. Data in **C** and **D** were from the same preparation as in *Figure 2A*.

probabilities p from 0.1 to 0.2. These parameter ranges encompass *n* = 325, an empirical estimate of the number of excitatory neurons in the perinatal mouse preBötC (*Hayes et al., 2012*) as well as p=0.13, the only experimentally determined connection probability among putatively rhythmogenic preBötC neurons in acute mouse slices (*Rekling et al., 2000*). Networks within the above *n*–p parameter range that generated respiratory-like cycle periods of ~4 s are shown with asterisks in *Figure 6A* and *Figure 6—figure supplement 1A*. This set of model networks also generated network-wide bursts within 200–300 ms following brief focal glutamatergic stimulation of five or more constituent neurons (*Figure 6B*) in agreement with focal glutamate un-caging experiments in neonatal mouse slices, which showed that simultaneous stimulation of 4–9 preBötC neurons can trigger inspiratory bursts with similar latency (*Kam et al., 2013b*). These results substantiate that the model networks well represent the neonatal mouse preBötC in vitro.

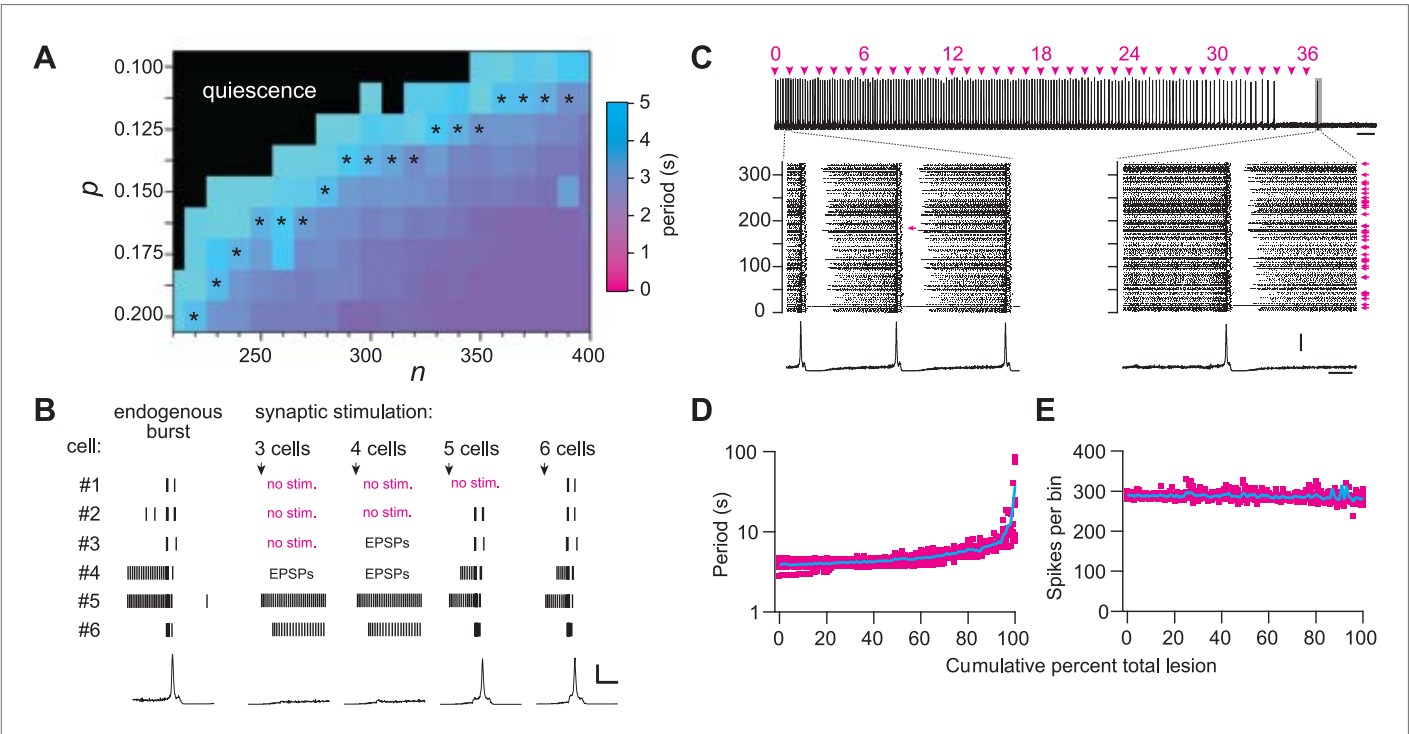

**Figure 6**. Numerical simulations. (**A**) Networks of Dbx1 preBötC neurons with population size (*n*) and synaptic connection probability (p). Blocks show the mean cycle period according to the colorimetric scale (right) for 10 (or more) realizations of the network for each (*n*,p) pair. Asterisks denote networks that generated respiratory-like (~4 s) cycle periods in ≥80% of individual realizations of the network. (**B**) Focal glutamatergic stimulation of constituent neurons in a model network (*n*,p) = (330,0.125). Network-wide bursts can be evoked when five or more individual cells are stimulated. These simulations mimic holographic laser-mediated glutamate un-caging experiments (*Kam et al., 2013b*) and are included because they bolster confidence that our model networks accurately capture features and behaviors of the preBötC in newborn mice. Raster plots show spike activity in six constituent neurons randomly selected from the network and focally stimulated (see 'Materials and methods' for numerical simulation of glutamate un-caging protocol). If focal stimulation evoked EPSPs (not spikes) then the raster reports 'EPSPs'; spikes are indicated by short vertical lines. From left to right, the number of stimulated units increments by one; five (or more) units evoked an inspiratory-like burst. A running-time histogram of network activity is shown at the bottom. Calibration bars represent 100 spikes/10-ms bin (vertical) and 0.5 s (horizontal). (**C**) Running-time histogram for one simulation of sequential ablation in a network (*n*,p) = (330, 0.125). Cell ablation tally is shown (top). Time calibration is 30 s. Spikes-per-bin calibration bar is the same as the inset (lower), 100 spikes/10-ms bin. Insets show a raster plot of spike activity in the entire network with a running-time histogram. The numerical y-axis reports cell index for each neuron model in the network. Left inset shows the first ablation (magenta arrow). Right inset shows all cumulative 36 ablations (magenta arrows). Time calibration for both insets is 1 s (at right). (**D**) Cycle period and (**E**) spikes-per bin (i.e., a measure of the magnitude of simulated network output as in **C**) are plotted vs cumulative percent of total ablations for 10 networks with (*n*,p) = (330, 0.125). **D** plotted in semi-log axes, **E** in linear axes. Magenta shows data from individual networks, cyan plots the mean response.

The following figure supplement is available for figure 6:

**Figure supplement 1**. Numerical simulations of Dbx1 neuron laser ablation experiments.

Sequentially deleting neurons in the model networks decreased frequency until the rhythm stopped altogether (*Figure 6C,D*, and *Figure 6—figure supplement 1B*; *Supplementary file 1*). These simulations generally agreed with the experimental results except for two discrepancies. First, the amplitude of network output did not diminish (compare *Figure 6C,E* to *Figures 2A and 3A*). Second, rhythm cessation required the average deletion of 41 constituent model neurons (SD 15, SEM 6, range 19–67, *Figure 6—figure supplement 1B,C*; *Supplementary file 1*) as opposed to the experimental cell ablation tally of ~85. First, we examine the loss of rhythmic function, and then address these discrepancies.

To assess whether a collapse of network structure could explain the breakdown in rhythmicity, we computed canonical local and global measures of topology for the graph G($n$,p) underlying each network simulation. These measures were computed after each cellular deletion and thus tracked continuously in parallel with the simulated networks. The table in *Supplementary file 2* reports the value of each topological measure prior to any deletions and after the final deletion associated with rhythm cessation (see 'Materials and methods' for definitions and computational methods). From the start to finish, the cumulative ablation sequence caused no major change in local metrics including cluster coefficient, closeness centrality, and betweenness centrality. Global connectivity metrics such as the K-core, which has been applied to analyze rhythmic neural systems including the preBötC (*Schwab et al., 2010*), showed only modest changes that were incommensurate with the large changes in frequency observed in experiments and simulations. The number of strongly connected components in the model networks did not depart from unity, thus the underlying graph was not fractured and every constituent interneuron could be reached via a finite number of synaptic links from every other interneuron, even after the rhythm stopped. These calculations show that these relatively low numbers of cumulative cellular ablations do not disconnect or disintegrate the core CPG, which suggests that a breakdown in network structure cannot explain the impairment and cessation of rhythmic function. The alternative is that neurons and synapses confer non-linear functional properties to the underlying rhythmic system that are not captured by the graph connectivity alone (see 'Discussion').

## Dbx1 preBötC neurons with premotor function

Lower cell ablation tallies perturbed and stopped the rhythm in simulations, and the aggregate burst magnitude did not decline (*Figure 6C,E*, and *Figure 6—figure supplement 1C*, *Supplementary file 1*). Both disparities could be explained if a subset of the experimentally lesioned population consists of premotor—rather than rhythmogenic—interneurons. Thus, we tested whether Dbx1 preBötC neurons project to the XII motor nucleus. Of the eight Dbx1 neurons with inspiratory modulation (*Figure 7A–D*), two could be antidromically activated by stimulation within the XII nucleus. *Figure 7* shows representative data from such a Dbx1 neuron whose XII-evoked antidromic spike was extinguished by collision with an orthodromic spike triggered by a somatic current pulse (*Figure 7E*). Most Dbx1 preBötC neurons are inspiratory and show commissural axons that cross the midline and innervate the contralateral preBötC (*Figure 8A–C*), as shown previously (*Bouvier et al., 2010*; *Picardo et al., 2013*). Here, we identify Dbx1 preBötC neurons that are also inspiratory modulated but send axons ipsilaterally toward the XII nucleus (*Figure 8D–F* and *Figure 8—figure supplement 1*), consistent with a role related to premotor transmission of inspiratory drive from preBötC to XII motoneurons.

## Discussion

Central pattern generators give rise to motor behaviors that are measurable in living animals as well as in reduced preparations that facilitate cellular-level investigations. To discover the core rhythmogenic and premotor interneurons that comprise CPGs and establish their functional parameters remain formidable problems, particularly in mammalian systems.

This report addresses the Dbx1 core hypothesis, which posits that Dbx1 preBötC interneurons constitute the rhythm-generating core for mammalian inspiratory rhythm. We affirm this hypothesis using cell-specific, cumulative laser ablation experiments, which also help quantify the size of the Dbx1 preBötC population needed to defend inspiratory rhythmogenic function. Additionally, we provide unanticipated new evidence that Dbx1 preBötC neurons also serve in a premotor capacity. While the existence of Dbx1 premotor neurons in the preBötC is not a straightforward prediction of the Dbx1 core hypothesis, it does not contradict it. We propose that an inspiratory rhythmogenic core population co-localizes with a subpopulation of premotor neurons, and both have the same developmental-genetic lineage related to Dbx1. These results help elucidate the rhythmogenic and premotor

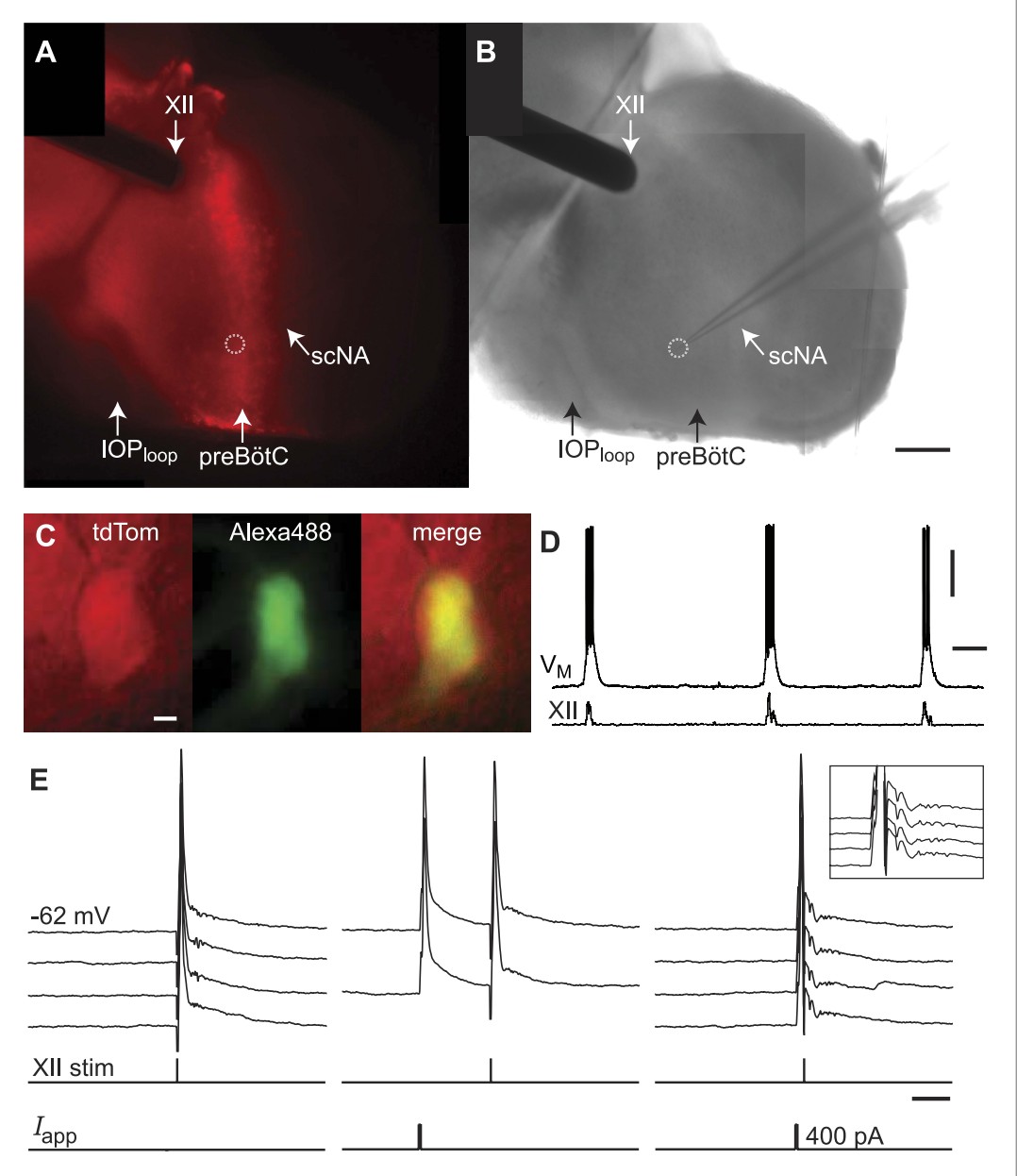

**Figure 7**. Dbx1 preBötC neurons with premotor function. (**A**) Fluorescence and (**B**) bright field images of a slice preparation. Anatomical landmarks are illustrated including: XII, the hypoglossal motor nucleus; scNA, semi-compact division of the nucleus ambiguus; IOP$_{loop}$, the ventral loop of the principal inferior olive, and the ventral surface of the preBötC. Scale bar is 100 µm and applies to **A** and **B**. A patch-recording pipette is visible, marking the inspiratory-modulated neuron detailed in **C–E**. A dotted circle indicates the tip of the pipette and cell body. (**C** and **D**) tdTomato expression, intracellular dialysis of Alexa 488, and merged image (**C**) from the inspiratory neuron shown with XII nerve output (**D**). Voltage and time calibration bars represent 20 mV and 1 s. Baseline membrane potential in the recorded neuron was −60 mV. (**E**) Antidromic activation of the Dbx1 inspiratory neuron from **C** and **D**. Action potentials were evoked by XII stimulation (left) and intracellular 5-ms supra-threshold current pulses (middle). When the antidromic XII stimulus was preceded immediately by a supra-threshold intracellular current pulse, the antidromic spike was occluded (collision test, right). Several sweeps, all from a −62 mV baseline membrane potential, are superimposed with vertical offset in each case. Voltage calibration is the same as panel **D**. Applied current ($I_{app}$) calibration is shown. Time calibration bar for **E** is 25 ms.

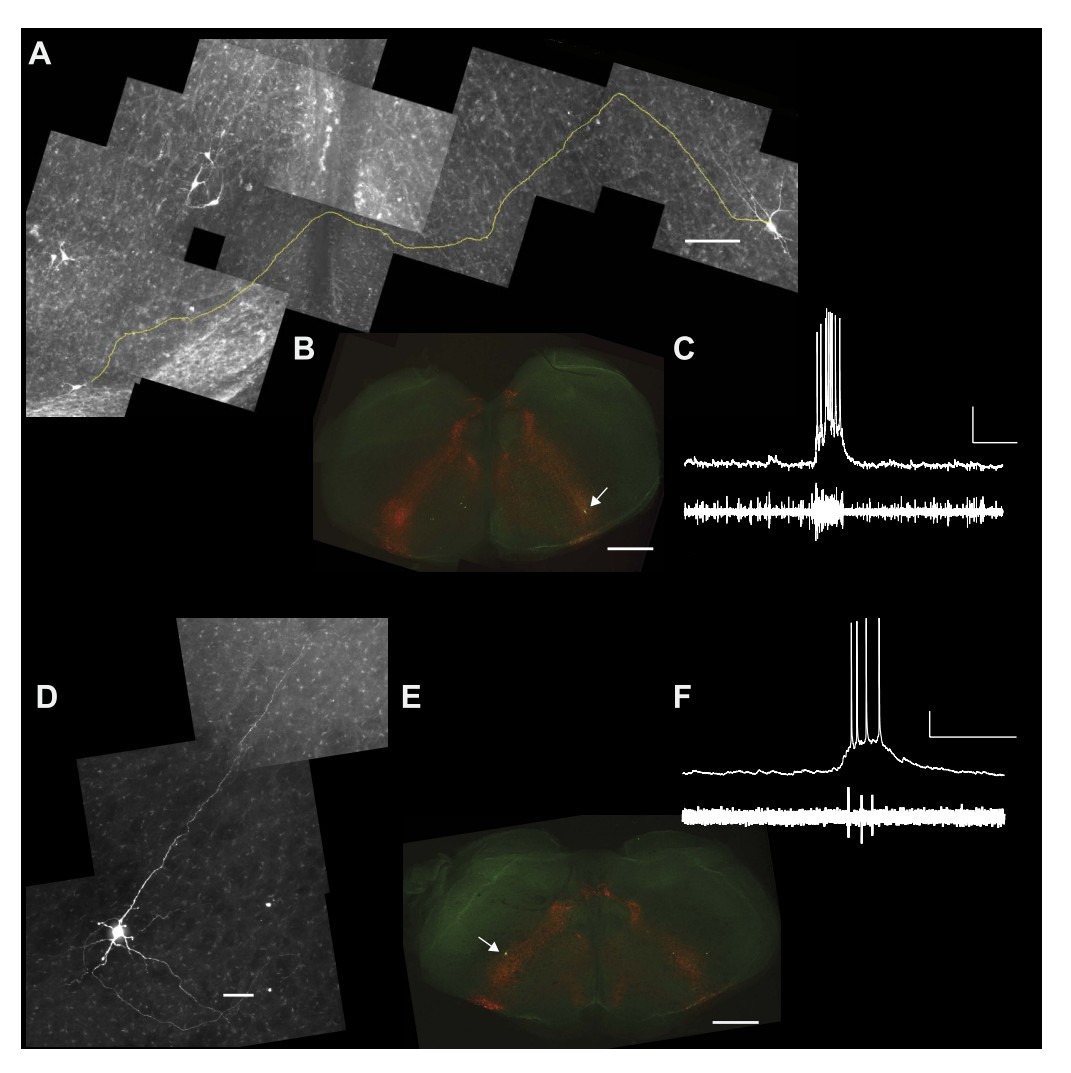

**Figure 8**. Commissural and premotor projections of inspiratory Dbx1 preBötC neurons. (**A**) Biocytin-filled and reconstructed Dbx1 preBötC neuron with commissural axon projection. The axon, which meanders in depth in this confocal image stack, was digitally traced (yellow) and superimposed in one plane for display. Axon trajectory crosses the midline of the slice and enters the preBötC contralaterally. Scale bar is 25 µm. (**B**) Mosaic image of the entire slice. The biocytin-filled soma (green) of neuron in **A** is shown at lower right (white arrow). Scale bar is 200 µm. Panels **A** and **B** have exactly the same orientation (dorsal up, ventral down). (**C**) Inspiratory discharge from the neuron in **A** and **B**. Top trace is membrane potential of the recorded Dbx1 preBötC neuron. Lower trace is XII output. Scale bars are 10 mV and 0.5 s. (**D**) Biocytin-filled and reconstructed Dbx1 preBötC neuron that projects toward the XII motor nucleus. Scale bar is 25 µm. The axon remained largely coplanar and thus is readily visible, except near its distal tip. In *Figure 8—figure supplement 1*, this same neuron is shown with a digitally traced (yellow) axon superimposed on the confocal image. (**E**) Mosaic image of the entire slice. Neuron in **D** is shown at lower left (white arrow). Scale bar is 200 µm. Panels **D** and **E** have exactly the same orientation (dorsal up, ventral down). (**F**) Inspiratory discharge from the neuron in **D** and **E**. Top trace is membrane potential of the recorded Dbx1 preBötC neuron. Lower trace is XII output. Scale bars are 10 mV and 0.5 s.

The following figure supplement is available for figure 8:

**Figure supplement 1**. Magnified view of the Dbx1 preBötC neuron from Figure 8D-F in which the axon has been digitally traced in the confocal stack and superimposed over the image to better illustrate the axon projection toward the XII motor nucleus.

components of a key mammalian CPG, and furthermore provide quantitative parameters that govern its functionality.

## Photonic interrogation of CPG structure and function

Two-photon lasers can destroy cells of a well-defined class with minimal damage to surrounding tissues (*Eklöf-Ljunggren et al., 2012*; *Wang et al., 2013*). Here, we use this technique to study the contribution of Dbx1 neurons in slices that capture essential components of the breathing CPG and generate measurable motor nerve output. Target detection relies on native fluorescent protein expression. An overwhelming majority of Dbx1 neurons in the ventral medulla have a glutamatergic transmitter phenotype and inspiratory modulated firing patterns (*Bouvier et al., 2010*; *Gray et al., 2010*; *Picardo et al., 2013*), so the Cre/lox Dbx1 reporter system is a reliable means to identify neurons with inspiratory function and target them for laser ablation.

*Dbx1* is also expressed in rostral parts of the ventral respiratory column between the caudal pole of the facial nucleus and the preBötC along the anterior–posterior axis (*Feldman et al., 2013*; *Gray, 2013*). The ventral respiratory column contains auxiliary inspiratory neurons (*Figure 4*) (*Smith et al., 1990*; *Ballanyi et al., 1999*; *Barnes et al., 2007*), which served as a control population. Laser ablating these neurons that do not have significant rhythmogenic function facilitates a comparative analysis of Dbx1 neuron ablations at the level of the preBötC. Laser ablation of Dbx1 neurons in the ventral respiratory column had a minor effect on XII motor output amplitude and negligible effects on frequency and regularity. These negative results show that laser–tissue interactions are not generally deleterious for respiratory function in vitro (*Eklöf-Ljunggren et al., 2012*; *Wang et al., 2013*).

## Dbx1 preBötC neurons form the inspiratory rhythmogenic core

Periodicity is the hallmark feature of an oscillator. Here, sequential laser ablation of Dbx1 preBötC neurons steadily diminished the inspiratory burst frequency, caused cycle period fluctuations, and ultimately the cessation of rhythmic motor output. We conclude that the oscillator was continuously degraded until it could no longer sustain spontaneous function. These data strengthen the proposal that Dbx1 neurons comprise the core inspiratory rhythm generator, which was originally based on *Dbx1* knockout mice that fail to breathe at birth (*Pierani et al., 2001*), and an array of neuroanatomical and physiological criteria including glutamatergic transmitter phenotype, the expression of peptides and peptide receptors, strong inspiratory rhythmic phenotype, and the ability to synchronize the preBötC bilaterally (*Bouvier et al., 2010*; *Gray et al., 2010*; *Picardo et al., 2013*).

We previously laser-ablated rhythmic preBötC neurons identified by $Ca^{2+}$ imaging. In that study, deleting all the detected targets (120 on average) slowed, destabilized, and then stopped the rhythm (*Hayes et al., 2012*). The interpretability of these experiments suffered two caveats: the rhythm stopped after a delay of ~30 min following the final target ablation, and furthermore, the transmitter phenotype of the ablated targets was unknown. In this study, use of the *Dbx1* Cre-driver line ensured that the target neurons were glutamatergic, a requisite characteristic for respiratory rhythmogenic function (*Greer et al., 1991*; *Funk et al., 1993*; *Ge and Feldman, 1998*; *Shao et al., 2003*; *Wallén-Mackenzie et al., 2006*). And here, destroying an average of 85 Dbx1 neurons stopped the XII motor rhythm in the midst of the ablation phase, before exhausting the target list, which suggests a more direct impact on the core rhythmogenic circuit. We cannot rule out the possibility that subsets of preBötC neurons generate 'burstlets' observable in local field recordings (*Kam et al., 2013a*). However, there is no collective inspiratory motor output after sequential laser ablation of Dbx1 preBötC interneurons, which indicates that the CPG is nonfunctional. Here, the debilitating effects on respiratory rhythm generation at a much lower ablation tally suggest that the preBötC core in vitro is very sensitive to the loss of just a few constituent interneurons (i.e., <100). This sensitivity to neuron loss may be accentuated in reduced slice preparations lacking excitatory and neuromodulatory drive from the rostral medulla and pons, as well as peripheral chemosensory and mechanosensory feedback via the vagus nerve. Extrinsic sources of drive raise preBötC network excitability and enhance respiratory rhythm. An acute sensitivity to neuron loss, such as we report for slices, may not apply to the preBötC network in vivo, but this remains to be tested via quantitative cellular ablation experiments with physiological monitoring in intact animal models.

## Parameters of the preBötC rhythmogenic core

We detected an average of 705 Dbx1 target cells in preBötC-surface slices, but we conclude that a significant number were non-rhythmogenic. Some fraction of the detected targets can be discounted

as *Dbx1*-derived non-rhythmogenic glia (**Gray et al., 2010**). However, more significantly, some fraction manifests premotor function. The present evidence for premotor function in Dbx1 preBötC neurons with verified inspiratory discharge patterns (e.g., **Figures 7 and 8D–F**) is consistent with large-scale pressure ejections of biocytin in the preBötC region of another strain of *Dbx1*-reporter mice (*Dbx1^{LacZ}* knock-in), which labeled many midline-crossing axons as well as axons projecting to the XII nucleus (**Bouvier et al., 2010**). Even 20 years ago it was recognized that a fraction of the excitatory neurons in the preBötC, and immediately dorsal to preBötC, had premotor functionality (**Funk et al., 1993**). Because laser ablations in preBötC-surface slices decreased XII magnitude (e.g., **Figures 2A and 3A**), we propose that a non-negligible fraction of the ablated Dbx1 neurons were inspiratory modulated but non-rhythmogenic, and most likely constitute XII premotor neurons (**Peever et al., 2002**; **Chamberlin et al., 2007**; **Koizumi et al., 2008**; **Volgin et al., 2008**). This scenario explains why there was a decline in XII motor output in experiments (deletion of Dbx1 premotor neurons causes motor output to decline) that was not mimicked by model simulations of a pure rhythmogenic circuit. It also explains why sequential ablations in simulations perturbed and stopped the rhythm at much lower cell ablation tallies; a significant fraction of the photonically ablated Dbx1 neurons were unrelated to rhythmogenesis per se.

Although we lack quantitative certainty, if we assume that each of the two caveats above (i.e., the existence of Dbx1 glia and premotor neurons) explains ~10% of the detected targets, then the size of the essential preBötC core would be $N = 705 - 2 [0.1 (705)] = 564$, which is remarkably close to the estimate of ~600 from adult rat studies that enumerated the population size based on NK1R expression in the preBötC (**Gray et al., 2001**, **1999**). In our previous laser ablation study, we estimated population size to be ~325 (**Hayes et al., 2012**), which probably underestimates the population size because incomplete fluorescent $Ca^{2+}$ dye loading in slices precludes the detection of a significant fraction of the rhythmogenic preBötC network.

## Physiological significance

Our data suggest that the preBötC contains rhythmogenic and premotor interneurons that both develop from *Dbx1*-expressing precursors. It is surprising that Dbx1 neurons play these two roles in respiration when the role of Dbx1 neurons in spinal locomotor systems seems to be coordinating left–right limb alternation at any speed (**Lanuza et al., 2004**; **Talpalar et al., 2013**) rather than rhythm generation or premotor transmission. In that regard, Dbx1 preBötC neurons appear to have more in common with excitatory Shox2 interneurons of the lumbar spinal cord (a subset of V2 interneurons), which contribute to locomotor rhythm generation and premotor circuits downstream of the rhythm-generating core (**Dougherty et al., 2013**).

The present measurements imply that destroying on average $\bar{X} = 85$ of $N = 564$ Dbx1 preBötC neurons (15%) precludes spontaneous respiratory motor rhythm in vitro. The mean and its 95% confidence intervals are expressed as follows: $\bar{X} \pm \left( Z_{\alpha/2} \dfrac{\sigma}{\sqrt{n}} \right)$, where $Z_{\alpha/2}$ is the cutoff value for a two-tailed normal distribution with probability $\alpha = 0.05$, and $\dfrac{\sigma}{\sqrt{n}}$ is standard error. Thus, we conclude the ability of Dbx1 preBötC neurons to spontaneously generate rhythmic respiratory motor output in slice preparations is sensitive to deletion of $85 \pm 1.96(20)$ Dbx1 interneurons, i.e., 8–22% of its core. Although it cannot function spontaneously post-lesion, the rhythmogenic preBötC core does not appear to be destroyed by piecewise lesioning. Peptide injections evoked irregular transient bursts in lesioned preBötC-surface slices. Also, measures of local and global connectivity in lesioned network models remained undiminished after cumulative cell deletions stopped and precluded rhythmic function. Therefore, we propose that preBötC function depends on non-trivial properties that emerge from non-linear synaptic and intrinsic membrane properties. Although such properties remain to be definitively determined, we advocate—and have explicitly modeled—a 'group pacemaker' rhythmogenic mechanism. In a group pacemaker, each constituent neuron forms recurrent connections with other constituent neurons in a network of finite size and amplifies excitatory drive via synaptically triggered inward currents (**Rekling et al., 1996**; **Rekling and Feldman, 1998**; **Rubin et al., 2009**). If that is a viable explanation for rhythmogenesis, then it could account for the loss of spontaneous function in the laser ablation context. Far before the cell ablation tally destroys the underlying network and its connectivity, the removal of each constituent neuron that contributes to rhythmic burst generation through its ability to amplify synaptic drive has a profound and deleterious effect on network functionality. The likelihood that arrhythmic respiratory networks retain considerable numbers of constituent neurons

and interconnectivity suggests that unraveling the cellular and synaptic mechanisms of rhythmogenesis and motor output could be exploited to restore functionality in lesioned slices and, to the extent that our observations apply in vivo, to develop clinical therapies that bolster respiratory function in pathological conditions of animal models or human patients.

Dbx1 respiratory neurons in the medulla represent excellent potential targets for pharmacological intervention or gene therapy to treat respiratory pathologies. Potentially enhancing premotor functionality in *Dbx1*-derived neurons could ameliorate obstructive sleep apnea. Boosting the function of rhythmogenic Dbx1 neurons may mitigate central apneas of prematurity as well as opiate respiratory depression. Treatment strategies aimed at rhythmogenic Dbx1 neurons may help overcome the effects of a reduced quantity or efficacy of neurons within the preBötC due to neurodegenerative diseases or aging (*Benarroch, 2003*; *Benarroch et al., 2003*; *Tsuboi et al., 2008*).

## Materials and methods

### Ethical approval

The Institutional Animal Care and Use Committee at The College of William & Mary, which ensures compliance with United States federal regulations concerning care and use of vertebrate animals in research, approved the following protocols. The anesthesia and surgery protocols are consistent with the 2011 guidelines of the Animal Research Advisory Committee, which is part of the Office of Animal Care and Use of the National Institutes of Health (Bethesda, MD).

### Animal models

We used transgenic mice that express Cre recombinase fused to the tamoxifen-sensitive estrogen receptor (CreERT2) in cells that express the *Dbx1* gene (*Dbx1$^{+/CreERT2}$*) (*Hirata et al., 2009*; *Gray et al., 2010*; *Picardo et al., 2013*). *Dbx1$^{+/CreERT2}$* mice were coupled to *floxed* reporter mice whose *Rosa26* locus was modified by targeted insertion of a *loxP*-flanked STOP cassette followed by tandem dimer (td) Tomato (*Gt(ROSA)26Sor$^{flox-stop-tdTomato}$*, i.e., *Rosa26$^{tdTomato}$*, Jax No. 007905) (*Madisen et al., 2010*). Tamoxifen administration to pregnant females on the tenth day after the plug date produces bright native fluorescence in *Dbx1*-derived neurons (i.e., Dbx1 neurons) in ~50% of the offspring: *Dbx1$^{+/CreERT2}$*; *Rosa26$^{tdTomato}$*. Dbx1 neurons can be visualized via native fluorescence in the preBötC and contiguous regions of the medulla. The *Dbx1$^{+/CreERT2}$* heterozygous line has a CD-1 background. The *Rosa26$^{tdTomato}$* line is homozygous with C57BL/6J background. We verified animal genotype via real-time PCR using primers specific for Cre and tandem dimer red fluorescent protein.

### Neonatal mouse slice preparations

Neonatal pups aged postnatal days 0–5 (P0–5) were anesthetized for at least 4 min of immersion in crushed ice in order to render the animals insentient to the same degree as would occur with inhalation anesthetics (*Danneman and Mandrell, 1997*; *Fox et al., 2007*). Anesthesia via hypothermia facilitates the rapid isolation of the intact brainstem and spinal cord, which would otherwise be damaged by cervical dislocation. The brainstem and spinal cord were removed within 90 s and then dissected in a dish filled with artificial cerebrospinal fluid containing (in mM): 124 NaCl, 3 KCl, 1.5 CaCl$_2$, 1 MgSO$_4$, 25 NaHCO$_3$, 0.5 NaH$_2$PO$_4$, and 30 D-glucose, equilibrated with 95% O$_2$ and 5% CO$_2$ (pH = 7.4). After removing the meninges and isolating the XII nerve rootlets, the brainstem and contiguous upper cervical spinal cord were fixed in position on a paraffin-coated paddle, or glued to an agar block, with rostral side up. The paddle or block was mounted to the vise of a vibrating microtome. The advancing blade approached the ventral surface of the tissue for sectioning in the transverse plane. XII nerve rootlets remained visible during the sectioning sequence. We cut a single slice of thickness 400–450 µm, which invariably retained the preBötC, XII premotor neurons and XII motoneurons that modulate and control airway resistance during breathing.

We employed two discrete slice-cutting strategies to differentially expose respiratory nuclei at the slice surface, as previously described (*Hayes et al., 2012*). The first slice type exposed the preBötC at the rostral face, and thus is called a *preBötC-surface slice*. The second slice type exposes the ventral respiratory column ~100 µm rostral to the preBötC at the rostral slice surface and served as a control slice for laser ablations. Histology atlases for newborn mice were used to calibrate slices online during sectioning (*Ruangkittisakul et al., 2011*, *2014*). For premotor recording experiments (*Figure 7*), we modified the preBötC-surface slice for the whole cell recordings such that the preBötC was exposed on the caudal surface.

Slices were perfused with 27°C ACSF at 4 ml/min in a recording chamber on a fixed stage upright microscope. The external $K^+$ concentration was raised to 9 mM and inspiratory motor output was recorded from XII nerve roots using a suction electrode and an AC-coupled differential amplifier. The amplified electrical signal and a root-mean-squared (smoothed) version of the signal were recorded by a 16-bit analog-to-digital converter and stored on a digital computer.

Because the composition of neural circuits at the rostral surface of the slice is critical for data interpretation, we fixed and stained each slice used for ablations at the end of the experiment to more precisely benchmark the neuroanatomical boundaries of respiratory-related nuclei according to the respiratory brainstem mouse atlases referred to above (*Ruangkittisakul et al., 2011, 2014*). Fixation solution contained 4% paraformaldehyde in phosphate buffer (33 mM $NaH_2PO_4$ and 67 mM $Na_2HPO_4$, pH = 7.2). After 1-hr in fixation solution, slices were rinsed in phosphate buffer for 2 min, and then submerged for 60–75 s in staining solution containing 1% thionin acetate, 0.1 M sodium acetate trihydrate, and 0.1 M acetic acid. After washing in a series of ethanol solutions, slices were mounted in a well slide, obliquely illuminated, and digitally imaged via stereomicroscope. Control slices were characterized by the compact division of the nucleus ambiguus (cNA), a thick dorsal inferior olive (IOD), and a minimally developed principal loop of the inferior olive ($IOP_{loop}$). The preBötC-surface slices were characterized by very little (if any) visible portion of the cNA yet a clear semi-compact nucleus ambiguus (scNA), a fully developed $IOP_{loop}$, as well as medial inferior olive (IOM).

## Electrophysiology

We performed whole cell recordings using a Dagan (Minneapolis, MN) IX2-700 current-clamp amplifier. Patch pipettes were fabricated from borosilicate glass (OD: 1.5 mm, ID: 0.87 mm, 4–6 MΩ in the bath) and filled with solution containing (in mM): 140 K-gluconate, 10 HEPES, 5 NaCl, 1 $MgCl_2$, 0.1 EGTA, 2 Mg-ATP, 0.3 Na-GTP, 50-µM Alexa 488 hydrazide, and 2-mg/ml biocytin. Empirical measurement of the liquid junction potential was 1 mV and thus not corrected. Access (series) resistance was ~10–15 MΩ, which was countered by bridge balance. Conventional current-clamp analog recordings were digitized at 4 kHz with a 16-bit A/D converter after 1 kHz low-pass filtering (PowerLab, AD Instruments, Colorado Springs, CO).

Neurons were selected for recording based on native tdTomato fluorescence in neurons preferentially in the dorsal preBötC. After identifying an inspiratory Dbx1 preBötC neuron, we tested for antidromic activation using a concentric bipolar electrode (FHC Inc., Bowdoin, ME) placed at the surface of the XII nucleus. Stimuli were triggered by a pulse generator (Tenma TGP110 10 MHz Pulse Generator, Aim-TTi USA, Fairport, NY) and amplitude and polarity were controlled by a stimulus isolation unit (Iso-Flex, AMPI, Jerusalem, Israel). We applied cathodic stimuli at increasing intensities, to a maximum of 0.4 mA, to elicit short latency antidromic action potentials. Then, brief (1 ms) current pulses, at magnitudes at or exceeding rheobase, were applied before the antidromic stimulation such that both ortho- and antidromic spikes were evoked. The delay between stimuli was progressively decreased until a collision was observed, i.e., the antidromic spike was occluded.

## Biocytin cell reconstruction

Biocytin-loaded neurons were fixed in 4% paraformaldehyde in 0.1 M Na-phosphate buffer for at least 16 hr at 4°C. Then, the slices were treated with Scale solution containing 4 M urea, 10% (mass/volume) glycerol and 0.1% (mass/volume) Triton X-100, for 10 days to clear the tissue and remove opaque background staining (*Hama et al., 2011*). Slices were then washed in phosphate buffered saline (PBS) for 1 hr, followed by a 15-min cycle with PBS containing 10% heat-inactivated fetal bovine sera (F4135; Sigma-Aldrich). Next, slices were incubated in PBS containing fetal bovine sera with additional 1% Triton X-100. Finally, the slices were incubated in FITC (i.e., fluorescein-isothiocyanate)-conjugated ExtrAvidin (E2761; Sigma-Aldrich) overnight at 4°C, and then rinsed twice with PBS, followed by six 20-min washes in PBS, and then cover-slipped in Vectashield (H-1400 Hard Set, Vector Laboratories, Burlingame, CA). We visualized recorded neurons using a laser-scanning confocal microscope (Zeiss LSM 510, Thornwood, NY) or a spinning-disk confocal microscope (Olympus BX51, Center Valley, PA). Images were contrast enhanced and pseudo-colored using the free *ImageJ* software (National Institutes of Health, Bethesda, MD), and then digitally reconstructed using the free *Neuromantic* software for morphological reconstruction (*Myatt et al., 2012*).

## Laser ablation: target detection

Dbx1 neurons were detected and mapped within three-dimensional (3D) volumes of the preBötC or ventral respiratory column, and then subsequently laser ablated while monitoring respiratory network functionality. The instrument incorporated a Zeiss LSM 510 laser scanning head and fixed-stage microscope body with a 20×/1.0 numerical aperture water-immersion objective, an adjustable wavelength 1.5 W Ti:sapphire tunable laser (Spectra Physics, Irvine, CA), and a robotic xy translation stage (Siskiyou Design, Grants Pass OR). The methodology has been described in a technical report (*Wang et al., 2013*) and in an original research report (*Hayes et al., 2012*).

We wrote custom software dubbed *Ablator* that automated a three-step routine. The first step (initialization phase) defines the domain for target detection and ablation. The domain can be bilaterally distributed, like the preBötC and ventral respiratory column. The maximum size of any part of the domain in the transverse (xy) plane must fit within an area of maximum dimensions 412 square micrometers. The z domain (depth) is a function of tissue opacity, laser power (Ti:sapphire), and the emission properties of the fluorescent reporter. For neonatal mouse brainstem tissue (P0–5), using 800-nm pulses emitted at ~1 W, which measured 36 mW at the specimen plane, the z domain generally measured less than 100 μm.

The second step (detection phase) acquires high-resolution images via confocal microscopy with a visible-wavelength laser (HeNe 543 nm for tdTomato). Dbx1 neurons were identified by native fluorescent protein expression using a threshold-crossing target detection algorithm in Ablator software, which is open-source and available for free download at the sourceforge.net archive, i.e., http://sourceforge.net/projects/ablator/. Additional image processing routines differentiate Dbx1 somata from auto-fluorescent debris and neuropil (*Figure 1—figure supplement 2*). The final map of Dbx1 neuron targets reflects the position of the center of each cell body in the 3D volume of the domain (see *Figure 1D*).

## Iterative threshold-crossing algorithm and image processing

Given an image that captures features of potential targets, Ablator calls the *Analyze Particles* routine in ImageJ, which is free image analysis software in the public domain (*Schneider et al., 2012*), to select ROIs. This routine detects particles using a threshold based on pixel intensity. It starts with a high value (near the maximum) and iteratively drops the threshold while accumulating particles (i.e., target ROIs) within a certain range of areas specified within the Ablator configuration. At first, with a high threshold, few local maxima are detected and the mask is small and sparse with ROIs. The routine then lowers the threshold by a user-defined increment and re-analyzes the image. As threshold decreases in steps, more ROIs become detectable. These newly detected ROIs are added to the mask, which expands the list of potential targets. The threshold is decreased incrementally over a number of partitions determined by the quotient of 4096 values of fluorescence intensity (for a 12-bit image) divided by the user-defined increment (above). As the detection process continues and threshold decreases incrementally, the smaller ROIs from prior iterations—which are fully contained in larger ROIs from the current iteration—are discarded, and the new larger ROIs are retained. Conversely, if a newly detected ROI at the current iteration envelopes two or more ROIs from a prior iteration, then the newly detected superset ROI is discarded and the multiple ROIs from the earlier iteration are retained. Thus the system avoids spuriously conflating two (or more) cells into a single target. After looping through all the partitions, the remaining set of ROIs is saved as the mask of potential target neurons for that focal plane.

## Circularity test

Ablator evaluates the circularity of ROIs as part of the threshold-crossing algorithm. A circularity score $C = 4\pi\frac{a}{p^2}$, is computed based on $a$ (area) and $p$ (perimeter) of the ROI. $a$ and $p$ are measured by the *Analyze Particles* routine in ImageJ (*Schneider et al., 2012*). $C$ ranges from 0 to 1. Scores near 0 denote an elongated polygon. $C$ approaches 1 for a perfect circle. If C falls below a user-specified cut-off, then the ROI is rejected from the target list. The appropriate C score depends on the characteristic morphology of the neurons of interest. Valid Dbx1 neurons in the preBötC and ventral respiratory column pass the circularity test when $C$ exceeds 0.75. Circularity is particularly useful in selecting somata rather than neuropil or auto-fluorescent detritus as valid targets (*Figure 1—figure supplement 2*). Rejecting an isolated dendrite segment (*Figure 1—figure supplement 2B,C*) avoids protracted lesion attempts during the ablation phase of the experiment, which are problematic because the dendrite and its soma are redundant targets and attempting to laser-ablate the dendrite is more likely to sever the process rather than kill the neuron (*Kole, 2011*).

## Priority rule

This final processing step eliminates redundant targets from adjacent focal planes. The same cell targets can be detected, and pass the circularity test, in more than one plane. When overlaying ROIs exist within adjacent focal planes the earliest acquisition from the deeper plane is retained and all other superficial ROIs are deleted (*Figure 1—figure supplement 2C,D*).

## Cell-specific laser ablation

Ablator chooses Dbx1 neuron targets in random order and advances until all the targets are exhausted or the respiratory rhythm ceases for longer than 120 s. The Ti:sapphire laser scans a 10 square micrometer spot centered on each target with 800-nm pulses at maximum intensity. The ablation is confirmed if fluorescence is detected in the band 560–615 nm, which reflects presumed water vapor in the cell cavity and excludes infrared reflections of the long-wavelength laser (*Figure 2—figure supplement 1A*) (*Wang et al., 2013*). In addition, lesioned targets disappear from the fluorescence image (*Figure 2—figure supplement 1B*), and their pre-lesion bright field image (*Figure 2—figure supplement 1C*) is replaced post-lesion by a pock mark (*Figure 2—figure supplement 1D*). Confirmed lesions add to a running tally. If lesion confirmation cannot be obtained, then the target selection algorithm does not advance and subsequent attempts are made to lesion the ROI. With each subsequent iteration, the scanning speed is decreased to improve the likelihood of lesioning the target. This loop repeats a total of five times. If confirmation of lesion cannot be ascertained after the fifth attempt, then it is deemed a failed lesion. Failed lesions do not contribute to the tally and their ROIs are removed from the list of targets to avoid reselection for the remainder of the experiment. A log file documents lesions by index number and time of confirmation. The XII rhythm is monitored and recorded continuously so its state can be directly correlated with the lesion tally in real time. Cell targets are destroyed in successful lesions so their effects are cumulative. The laser lesions are performed bilaterally in the preBötC. After a batch of lesions on one side, the robotic xy translation stage translates to the contralateral side and performs another batch, and then switches sides again, and so on until the targets are exhausted or the XII rhythm ceases.

We measured XII burst magnitude (amplitude and area) and computed cycle period (the interval between consecutive XII bursts) using LabChart software (ADInstruments, Colorado Springs, CO). The regularity score (*RS*) was defined as the quotient of period of the present cycle $T_n$ with respect to the mean cycle period for ten previous cycles:

$$RS = \frac{T_n}{\frac{1}{j}\sum_{i=1}^{j} T_{n-i}},$$

where $j = 10$. We defined the control epoch as 30 min of continuous recording from the end of the detection phase to the beginning of the ablation phase. Data sets were tested for normality using a Shapiro–Wilk test. We rejected the null hypothesis that the data are drawn from a normal distribution if the p-value of the test statistic was less than $\alpha = 0.05$. Data that could be considered normally distributed were compared using two-tailed paired t-tests, whereas data that did not conform to the normal distribution were compared using non-directional (two-tailed) Mann–Whitney *U*-tests. XII burst amplitude and frequency/cycle period were reported with standard deviation (SD) and standard error of the mean (SEM). Discrete cell counts that pertain to the number of neurons detected or the number of neurons lesioned are reported with SD, SEM and min–max range.

## Network simulations and modeling

We wrote a Matlab (MathWorks Inc., Natick, MA) script to generate Erdős-Rényi G(*n*,p)-directed random graphs (*Newman et al., 2006*) with key parameters of population size (*n*) and connection probability (p). Vertices (a.k.a., nodes) of G(*n*,p) were populated by Rubin–Hayes preBötC neuron models and the directed edges (a.k.a., links) between vertices were modeled by excitatory glutamatergic synapses (*Rubin et al., 2009*). We simulated the network models on the SciClone computing complex at The College of William & Mary, which features 193 nodes with a total of 943 CPU (central processing unit) cores, 5.9 terabytes of physical memory, 220 terabytes of disk capacity, and peak performance of 21.2 teraflops. We used a Runge–Kutta fourth-order numerical integration routine with fixed time step of 0.25 ms. Network models were subject to 100 random deletions, one deletion every 25 s. Neuron deletions were achieved by setting the synaptic state variable and its corresponding differential equation to zero,

which essentially removes the cell from the network. Deleted neurons no longer contributed to running-time histograms of network activity and were removed from raster plots (e.g., *Figure 6C*). Transient glutamatergic stimulation of constituent model neurons mimicked the experimental gluta-mate un-caging protocol by Kam et al. which evoked respiratory bursts in the preBötC (*Kam et al., 2013b*). Focal stimulation was achieved by setting the synaptic state variable to 0.9 for 200 ms, without modifying the differential equation, so the glutamatergic excitation was indeed transient. Focal stimulation was applied to rhythmically active networks several seconds following an endoge-nous burst (*Figure 6B*).

Since there is uncertainty regarding the exact network size, we conducted a series of simulations for a range of ($n$,p) with the aim of finding a reasonable parameter range to produce respiratory-like rhythms (3-4 s cycle period prior to ablations). We varied $n$ from 200 to 400 with a step size of 10 and p from 0.1 to 0.2 with a step size of 0.0125. For each parameter set, 10 simulations without deletion were conducted for 25 s to assess network rhythmicity (*Figure 6—figure supplement 1*).

For the parameter sets whose initial period fell between 3 and 4 s, we performed 5–6 simulations with deletions (for $n$ = 320, 330, 340 we performed 16 simulations in each case) and then calculated the longest period, the ablation tally, and discrete network metrics pertaining to G($n$,p). The results are documented in *Figure 6* and *Figure 6—figure supplement 1*, as well as the table in *Supplementary file 1*. During the simulations, raster graphs were simultaneously generated to detect the spiking for each individual neuron (*Figure 6C*). The running time histogram is based on the raster graph for each simulation, from which we computed the cycle period and amplitude (number of spikes per time bin, *Figure 6D,E*).

## Discrete network simulations

A network with $n$ vertices can be represented by its adjacency matrix $A(n \times n)$ in a manner that if there is a connection from vertex $i$ to vertex $j$ then $A_{ij}$ = 1, otherwise $A_{ij}$ = 0. The adjacency matrices are asymmetric for neuronal networks, which are directed (i.e., the chemical synapses are unidirectional). In discrete simulations, the lesion of neurons is modeled by removing vertices from the adjacency matrix along with their edges, i.e., connections (in and out). We computed three global metrics (K-core, number of strongly connected components, average in and out degree) for the initial network and the remaining network after a sequence of 100 random deletions. Also for each deleted vertex, we computed three local network metrics (local cluster coefficient, closeness centrality, betweenness cen-trality) to indicate the importance of the vertex within the previous network. The metrics are defined below and reported in the table of *Supplementary file 2*.

### K-core
It refers to the maximum sub-graph such that each vertex of the sub-graph has at least K edges (connections). In this case, an in-arc and an out-arc both count as an edge.

### Strongly connected components (SCC)
The strongly connected components of a directed graph G($n$,p) are its maximal strongly connected sub-graphs, such that within each sub-graph there is a path from each vertex to every other vertex. Therefore the number of SCC can exceed unity. Nonetheless, when SCC = 1 the existing network is said to be fully connected, i.e., there are no isolated islands and every vertex can connect to every other vertex via a finite number of edges.

### Average in and out degree
In an $n \times n$ adjacency matrix $A$ of a directed graph G($n$,p), $A_{ij}$ = 1 refers to a connection from vertex $i$ to $j$, and $A_{ji}$ = 1 refers to a connection from vertex $j$ to $i$. Therefore $\sum_{j=1}^{n} A_{ij}$ is the out-degree for node $i$ while $\sum_{j=1}^{n} A_{ji}$ is the in-degree.

### Local cluster coefficient
Local cluster coefficient measures how close the neighbors of the vertex are to being a complete graph, i.e., a graph where each vertex is connected to every other vertex. For a vertex $v_i$ with $k_i$ edges, the local cluster coefficient is defined as

$$C_i = \frac{|\{A_{jk} : v_j, v_k \in N_i, A_{jk} = 1\}|}{k_i(k_i - 1)}$$

where $N_i$ is the neighborhood of $v_i$, the sub-graph formed by all the vertices $v_i$ connects to (that is, all the out-neighbors of $v_i$). The numerator is the number of actual connections within $N_i$ while the denominator is the number of connections if $N_i$ is a complete graph.

## Closeness centrality

For a vertex $v_i$, the farness is defined to be the sum of shortest paths from $v_i$ to every other reachable vertex. The closeness of $v_i$ is the inverse of the farness. The closeness centrality for $v_i$ is defined as the product of the number of vertices in the graph $n$ and the closeness of $v_i$. From this definition, a central vertex would have a small farness and a large closeness centrality.

## Betweenness centrality

Betweenness centrality measures the frequency that a vertex acts as a bridge in the shortest path between two other vertices. It is defined as $C_B(v) = \sum_{s \neq v \neq t} \frac{\sigma_{st}(v)}{\sigma_{st}}$ where $s$, $v$, $t$ are three different vertices in the graph, and $\sigma_{st}(v)$ is the number of shortest paths between $s$ and $t$ through $v$, while $\sigma_{st}$ is the total number of shortest paths between $s$ and $t$. Betweenness centrality is usually normalized by dividing the number of total possible vertex pairs $(n-1)(n-2)$, excluding $v$.

## Acknowledgements

Funding sources: NIH grants HL104127 and NS070056 (PI: Del Negro) and CIHR grant RES0018140 (PI: Funk). Computational facilities at the College of William and Mary were supported by the National Science Foundation, the Virginia Port Authority, Sun Microsystems, and Virginia's Commonwealth Technology Research Fund.

## Additional information

### Funding

| Funder | Grant reference number | Author |
|---|---|---|
| National Heart, Lung, and Blood Institute | HL104127 | Hanbing Song, Andrew Kottick, Nikolas C Vann, Maria Cristina D Picardo, Victoria T Akins, Christopher A Del Negro |
| National Institute of Neurological Disorders and Stroke | NS070056 | Xueying Wang, John A Hayes, Christopher A Del Negro |
| Canadian Institutes of Health Research | RES0018140 | Ann L Revill, Gregory D Funk |

The funders had no role in study design, data collection and interpretation, or the decision to submit the work for publication.

### Author contributions

XW, JAH, CADN, Conception and design, Acquisition of data, Analysis and interpretation of data, Drafting or revising the article; ALR, HS, AK, NCV, MCDP, VTA, Acquisition of data, Analysis and interpretation of data; MDLM, Conception and design, Acquisition of data, Analysis and interpretation of data; GDF, Acquisition of data, Analysis and interpretation of data, Drafting or revising the article

### Author ORCIDs

Xueying Wang, http://orcid.org/0000-0002-2399-8083
John A Hayes, http://orcid.org/0000-0003-2849-7672
Ann L Revill, http://orcid.org/0000-0002-6071-6866
Hanbing Song, http://orcid.org/0000-0002-2164-8813
Andrew Kottick, http://orcid.org/0000-0002-3731-5140
Nikolas C Vann, http://orcid.org/0000-0003-4139-0642
M Drew LaMar, http://orcid.org/0000-0002-4037-1848

Maria Cristina D Picardo, http://orcid.org/0000-0001-8912-2175
Victoria T Akins, http://orcid.org/0000-0002-0061-7612
Gregory D Funk, http://orcid.org/0000-0001-5848-0631
Christopher A Del Negro, http://orcid.org/0000-0002-7848-8224

## Ethics

Animal experimentation: The Institutional Animal Care and Use Committee (IACUC) at The College of William & Mary, which ensures compliance with United States federal regulations concerning care and use of vertebrate animals in research, approved the following protocols (IACUC-2013-07-10-8828-cadeln). The anesthesia and surgery protocols are consistent with the 2011 guidelines of the Animal Research Advisory Committee, which is part of the Office of Animal Care and Use of the National Institutes of Health of the USA.

## Additional files

### Supplementary files

• Supplementary file 1. Numerical simulations of Dbx1 neurons in model preBötC networks subjected to cumulative laser ablation experiments. Erdős-Rényi random directed graphs G($n$,p) were populated with Rubin-Hayes preBötC neuron models at each node, and their links were described by excitatory synapses, as described for *Figure 6* and *Figure 6—figure supplement 1* above. In numerical simulations, the resulting network models with very high probability of generating rhythm and respiratory-like cycle period (~4 s, indicated by asterisks in *Figure 6A* and *Figure 6—figure supplement 1*) were subjected to piecewise cumulative ablation protocols like slice experiments (*Figures 2–4*). The parameters describing the model networks (number of neurons $n$ and synaptic connection probability p) are listed below in columns 1 and 2. Each ablation experiment was simulated five or more times. The maximum period (in s) and cumulative ablation tally (unitless) required to stop the rhythm are listed in the table for each individual realization of the network model along with average values for these characteristic measures. The networks deemed to be most representative of the preBötC, i.e., ($n$,p) = (320, 0.1375), (330, 0.125), (340, 0.125) were simulated 16 times each.

• Supplementary file 2. Discrete network simulations. As in the main text and the table above, parameters ($n$,p) represent the number of constituent neurons and connection probability. Here the networks are Erdős-Rényi static directed random graphs G($n$,p) (as in *Figure 6* and *Figure 6—supplement 1*); that is, the nodes are not populated with dynamical models and the interconnections between nodes are simply static directed links (rather than dynamical synapses). Each network (static graph) below was subjected to 100 random deletions. During the ablation sequence, we computed the following global network metrics: K-core, the number of strongly connected components (SCC), and local network metrics: local cluster coefficient, closeness centrality, and betweenness centrality. The initial (first) and final (last) values for the global and local measures are plotted side by side in the appropriate columns below. We also computed the initial in- and out-degree (i.e., the average number of directed connections in and out), percentage drop in the final average in-degree, and the percentage-drop of the final average out-degree. The 'Ave.' row reports average change (in percent) for K-core, SCC, cluster coefficient, closeness centrality, and betweenness centrality, as well as the average in- and out-degree for initial and final states of the network. Definitions for the characteristic measures are elaborated in the 'Materials and methods'.

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
