## [Decision Letter]

Thank you for sending your work entitled “Laser ablation of Dbx1 pre-Bötzinger Complex interneurons abolishes inspiratory rhythm and impairs pattern” for consideration at *eLife*. Your article has been favorably evaluated by Eve Marder (Senior editor) and 3 reviewers, one of whom, Ronald L Calabrese, is a member of our Board of Reviewing Editors.

The Reviewing editor and the other reviewers discussed their comments before we reached this decision, and the Reviewing editor has assembled the following comments to help you prepare a revised submission.

The authors combine molecular-genetic, laser ablation, electrophysiological and computational experiments to determine whether Dbx1 glutamatergic neurons of the neonatal mouse pre-Bötzinger Complex represent the rhythmogenic core of the respiratory CPG. They show elegantly that they do indeed and that progressive removal of just a small percentage of the population stops rhythmicity (which can then be transiently restored by SP injection) confirming a long standing hypothesis. More surprisingly ablation of these neurons diminishes motor outflow, suggesting that some Dbx1 interneurons are premotor. They then show with a few examples that in contrast to the rhythmogenic interneurons, which project to the contralateral pre-Bot, interneurons in the Dbx1 population project ipsilaterally to the motor nucleus XII suggesting that they are premotor. Computational modeling suggests that a proportion of the Dbx1 interneurons are not involved in rhythmogenesis.

This paper is clearly written and well documented with convincing data. Each figure conveys important data. This work is a significant and necessary step forward from previous laser ablation studies by the same lab that identified potential rhythmogenic targets by Ca imaging [Hayes JA, Wang X, Del Negro CA. 2012. Cumulative lesioning of respiratory interneurons disrupts and precludes motor rhythms in vitro. PNAS 109(21): 8286-91].

There are concerns that should be addressed in revision by careful rewriting and with appropriate cautions on some of the conclusions.

1) A major concern is how the authors seem to extrapolate the findings on the numbers of neurons that when ablated lead to rhythm termination to the *in vivo* state. The conclusion that the “preBötC core is very sensitive to just a few constituent neurons (i.e. <100)” may not be valid for the *in vivo* state of the preBötC network, a caveat that the authors should discuss in the text. Their own results with SP injection suggests that more excitation and/or modulation *in vivo* may make the rhythm generator immune to loss of 15% of the Dbx1 Interneurons. They should also draw back on their extrapolation of how these results relate to the loss of respiratory function in disease sates *in vivo*.

2) The effects deletions of Dbx1 neurons on motor discharge appear to be only on amplitude and not on patterning of the motor outflow. By pattern the reviewers mean phase relations of MNs not just the number of MNs activated or their spike frequency. Careful rewriting should make this distinction clear. The authors do not demonstrate experimentally that these neurons are NOT part of the rhythm-generating kernel of excitatory neurons and hence are “non-rhythmogenic” Dbx1 premotoneurons or directly that they are premotor. This inference/hypothesis came from the excitatory network model. It is a reasonable hypothesis but at present we don't know enough about the spatial organization and projection patterns of these Dbx1 neurons. It is possible that some of these neurons have axon collateral projections within the preBötC and also contralaterally in addition to ipsilateral projections to the hypoglossal motor nucleus. Also the authors have not shown us that the 2 cells that they identified actually have synaptic connections with motor neurons.

3) We have appended the original reviews because they emphasize different aspects of the paper but are non-conflicting. They can be fruitfully used to revise the manuscript.

Reviewer #1

The authors combine molecular-genetic, laser ablation, electrophysiological and computational experiments to determine whether Dbx1 glutamatergic neurons of the neonatal mouse pre-Bötzinger Complex represent the rhythmogenic core of the respiratory CPG. They show elegantly that they do indeed and that progressive removal of just a small percentage of the population stops rhythmicity (which can then be transiently restored by SP injection) confirming a long standing hypothesis. More surprisingly ablation of these neurons diminishes motor outflow, suggesting that some Dbx1 interneurons are premotor. They then show directly with a few examples that such premotor interneurons are in the Dbx1 population and that in contrast to the rhythmogenic interneurons, which project to the contralateral pre-Bot, they project ipsilaterally to the motor nucleus XII. Computational modeling corroborates that a proportion of the Dbx1 interneurons are not involved in rhythmogenesis.

This paper is clearly written and meticulously documented with convincing data. Each figure conveys important data and the Discussion appropriately points out potential clinical implications. This work is a significant and necessary step forward from previous laser ablation studies by the same lab that identified potential rhythmogenic targets by Ca imaging [Hayes JA, Wang X, Del Negro CA. 2012. Cumulative lesioning of respiratory interneurons disrupts and precludes motor rhythms in vitro. PNAS 109(21): 8286-91.].

Reviewer #2

This is a technically elegant study testing the hypothesis that neurons developmentally expressing the Dbx1 transcription factor in the pre-Bötzinger complex (preBötC) constitute the core component of the inspiratory rhythm generator in rhythmically-active neonatal mouse slices obtained from Dbx1-Cre-tdtomato transgenic mice. While it has been established that the preBötC is the critical structure for inspiratory rhythm generation, the precise identity of the preBötC excitatory interneurons generating the inspiratory rhythm has not been established. This study therefore represents an important attempt to establish that neurons derived from Dbx1-expressing precursors constitute a critical glass of interneurons and also establish that Dbx1 expression at the level of the preBötC is a reliable marker for the rhythm generating neurons. Previous studies have suggested that Dbx1-expressing neurons are a superset of neurons in the mouse central nervous system and in respiratory-related regions of the medulla including premotor regions are largely glutamatergic neurons and in the preBötC express several phenotypes that are thought to be hallmarks of inspiratory rhythm-generating interneurons. Thus it has been hypothesized in the field that Dbx1-expression may be a reliable surrogate marker for the critical population of glutamatergic preBötC neurons.

The authors have effectively employed a previously developed laser-based ablation system to cumulatively photoablate the Dbx1-tdTomato neurons one neuron at a time, which perturbed the inspiratory rhythm, monitored by recording the hypoglossal motor output from the slice preparation, and surprisingly completely disrupted rhythm generation after ablating only ∼15% of the estimated total Dbx1-expressing neuronal population in the preBötC, from which the authors suggest that preBötC rhythm generating core is very sensitive to loss of just a few constituent interneurons. The authors also report reductions in the amplitude of the motor output, consistent with previous findings that a subpopulation of preBötC inspiratory neurons have axonal projections directly to the XII motor nucleus and verified here for Dbx1-expressing neurons, which suggests a premotor pattern formation function of a subset of preBötC neurons. Simulations with a particular model of preBötC excitatory rhythmogenic circuits (Rubin-Hayes model) predict only perturbations of network burst frequency, consistent with the idea that non-rhythmogenic inspiratory preBötC neurons may be subserving a premotor pattern formation function.

One major concern to be addressed is that the quantification of the size of the Dbx1 preBötC population required to maintain rhythmogenic function *in vitro* (∼15% of the total Dbx1 population) may not be representative of the situation operating in vivo because of major differences in the level of Dbx1 neuron excitation and hence ability of this network population to generate inspiratory rhythm in different states. There are sources of excitatory/neuromodulatory drive to the preBötC operating *in vivo* (e.g. RTN) that are absent in the slice preparations and the slices are in fact in a low excitation state (i.e., compared to the *in vivo* mouse, very low inspiratory discharge frequency dependent on elevated extracellular potassium for stable slow rhythm generation). The conclusion that the “preBötC core is very sensitive to just a few constituent neurons (i.e. <100)” (and the authors suggestion that this may have ramifications for the loss of respiratory function in disease states) may not be valid for the *in vivo* state of the preBötC network, a caveat that the authors should discuss in the text. Have the authors effectively varied the state of excitation for example in the model network to gain insight into how level of excitation affects the critical mass of neurons required to sustain rhythmogenic function?

The results with the excitatory network model are instructive regarding perturbations of population burst frequency vs. amplitude with simulated single-cell ablation. The authors state (p. 8, para. 1) that some of their test simulation results substantiate that the model networks well represent the neonatal mouse preBötC in vitro.” In the experiments of Kam et al. 2013 cited, network bursts were generated after a substantial delay (200 - 300 ms) from the onset of neuronal activation. Does the present network model predict this temporal feature? Also the cellular burst generation and terminations mechanisms represented in the Rubin-Hayes model have not been verified experimentally, which should be noted in the Discussion. On the other hand, it could be that the type of test results used to make the claim about the model as a good representation may not be particularly sensitive to the actual burst generation and termination mechanisms incorporated.

An experimental issue to be addressed is that there is no information provided for the five slices studied in ablation experiments on the number of neurons ablated on each side of the slice. In the Materials and methods the authors state that “The domain can be bilaterally distributed like the preBötC and ventral respiratory column” and bilateral target maps are shown, but the authors should state how the targets were actually distributed in the experiments and any potential impact on the ablation results should be discussed.

Reviewer #3

This manuscript by Wang and colleagues from Del Negro's and Funk's labs reports on the role of Dbx-derived preBotzinger complex neurons in respiratory rhythm and pattern generation. They used a method to selectively and progressively ablate neurons in a slice preparation obtained from a transgenic mouse line in which cells of interest (Dbx-derived cells) are constitutively expressing a fluorescent marker (used to target these cells for the ablation procedure). They aimed to determine the impact of these ablations on network function in order to establish some quantitative cellular parameters controlling network functionality and test their “Dbx1 core hypothesis”. As predicted they found that after a progressive decrease in respiratory frequency (monitored through the recording of the XII nerve output), ablation of ∼85 cells led to the complete rhythm cessation. This effect was accompanied by a decrease in the motor output amplitude, suggesting a role of Dbx1-derived neurons in pattern formation.

While the paper is clearly written, easy to follow and the figures explicit I have a few concerns:

First the degree of novelty: the elegant method consisting in lesioning specific neuronal populations within a network and more specifically the respiratory network of the preBotzinger complex while monitoring the resulting motor output has been already presented (including all the methodological controls), (Hayes, 2012). And the important role of Dbx1-derived cells in constituting the core of the preBotzinger respiratory network has also been recently published (6; 20). Therefore the experiments dedicated to examine the effects of laser ablation of Dbx-derived respiratory neurons on the respiratory network function seem to only complement/confirm previous already published results, but do not really bring any novel results susceptible to significantly improve the current knowledge on respiratory rhythmogenesis.

Second I have a problem with the use of the word “pattern”. This might be only a semantic problem but it should be made clear what the authors mean with pattern. To my point of view “pattern” means the respective and relative organization of the different phases of a rhythmic activity. For example for the respiratory behavior this could include expiration vs inspiration phases, the distinction between the different sub-phases of the inspiration activity.... But, to me, the amplitude of the hypoglossal motor output recorded from a slice preparation is not a parameter that can be associated with patterning. If we agree on this, then part of the conclusions of the paper are incorrect and should be substantially revised.

In the Introduction the authors claim that “rhythmogenic preBotzinger neurons are distinguished by glutamatergic neurotransmitter” and that “GABA- and glycinergic neurons in the preBotC influence sensory integration but are non-rhythmogenic because blocking inhibition does not prevent the respiratory rhythm to be generated”. This is not completely true as Morgado-Valle (2010) provided evidence that some rhythmically active glycinergic neurons of the preBotC exhibit pacemaker properties and Winter (2009) showed that glycinergic neurons are significantly integrated in the preBotC network as they constitute half of the rhythmically active neuron in this network. The authors should at least site these work here.

In the legend of Figure 1 and Figure 4 it is written that the red dots in panel C correspond to Dbx1- neurons and the blue ones to non-Dbx1 neurons. How does the system detect the non-Dbx1 neurons? I understood that the cell to be deleted were selected on the basis of fluorescent protein expression. By definition the non-Dbx1 neurons should not be detected because they are not tdTom positive. Thus how can they be identified by the system? Please clarify this.

It is stated that the fluorescent neurons were preferentially selected in the dorsal part of the preBotC. Why this part more specifically? If similar experiments are performed exclusively in the ventral part of the preBotC is the ablation-induced effect on XII motor output amplitude still observed? Indeed Funk (1993), Koizumi (2013) and the present work locate the inspiratory premotoneurons in the dorsal aspect of the preBotC region, so one would expect that when ablations are performed in the ventral region of the preBotC then the consequences on motor output should be less. Did the authors ever observed a difference on results obtained depending on the ventral/dorsal position of the ablation domain? Such experiments would validate part of the results presented here and would be in complete agreement with the model used in this study.

---

## [Author Response]

*1) A major concern is how the authors seem to extrapolate the findings on the numbers of neurons that when ablated lead to rhythm termination to the in vivo state. The conclusion that the “preBötC core is very sensitive to just a few constituent neurons (i.e. <100)” may not be valid for the in vivo state of the preBötC network, a caveat that the authors should discuss in the text. Their own results with SP injection suggests that more excitation and/or modulation in vivo may make the rhythm generator immune to loss of 15% of the Dbx1 Interneurons. They should also draw back on their extrapolation of how these results relate to the loss of respiratory function in disease sates in vivo*.

We addressed this critique in response to Reviewer #2 below. The most important revisions to the text, which pertain to this critique, are now found in the Discussion section.

*2) The effects deletions of Dbx1 neurons on motor discharge appear to be only on amplitude and not on patterning of the motor outflow. By pattern the reviewers mean phase relations of MNs not just the number of MNs activated or their spike frequency. Careful rewriting should make this distinction clear. The authors do not demonstrate experimentally that these neurons are NOT part of the rhythm-generating kernel of excitatory neurons and hence are “non-rhythmogenic” Dbx1 premotoneurons or directly that they are premotor. This inference/hypothesis came from the excitatory network model. It is a reasonable hypothesis but at present we don't know enough about the spatial organization and projection patterns of these Dbx1 neurons. It is possible that some of these neurons have axon collateral projections within the preBötC and also contralaterally in addition to ipsilateral projections to the hypoglossal motor nucleus. Also the authors have not shown us that the 2 cells that they identified actually have synaptic connections with motor neurons*.

The Reviewing editor is correct. We did not demonstrate that Dbx1 preBötC neurons with premotor functionality are *not* also part of the rhythmogenic core, but the model-experiment disparity suggests that this is the case. That is, we had to delete approximately twice as many cells in the experiments as in the simulations of a purely rhythmogenic network to cause rhythm cessation. If some Dbx1 neurons were simultaneously part of the rhythmogenic and premotor subsets, then two predictions would arise: 1) we would expect the experiments to decline in output amplitude, but not the simulations (a pure rhythmogenic setup), and 2) we would expect both simulations and experiments to require similar ablation tallies to stop the rhythm. Prediction 2 was clearly not observed.

It is also correct that we did not demonstrate synaptic connections with XII motoneurons. Our evidence for premotor function is based on: i) a decline in XII output after laser ablation, ii) antidromic activation from the XII nucleus. (Note, the XII nucleus is virtually homogenous containing >95% motoneurons and only 5% GABAergic interneurons. Thus, it is highly probable that any axons antidromically activated via stimulation within the XII nucleus synapse on XII motoneurons.) And, iii) biocytin reconstruction showing axon trajectory toward/into the XII nucleus. These are indirect demonstrations, but taken together make a strong case in our view.

We further address this critique of ‘pattern’ in response to Reviewers #1 and #3, below.

*3) We have appended the original reviews because they emphasize different aspects of the paper but are non-conflicting. They can be fruitfully used to revise the manuscript*.

Reviewer #1

*[…] This paper is clearly written and meticulously documented with convincing data. Each figure conveys important data and the Discussion appropriately points out potential clinical implications. This work is a significant and necessary step forward from previous laser ablation studies by the same lab that identified potential rhythmogenic targets by Ca imaging [Hayes JA, Wang X, Del Negro CA. 2012. Cumulative lesioning of respiratory interneurons disrupts and precludes motor rhythms in vitro. PNAS 109(21): 8286-91]*.

We thank Reviewer #1 for a supportive assessment.

Reviewer #2

*[…] One major concern to be addressed is that the quantification of the size of the Dbx1 preBötC population required to maintain rhythmogenic function in vitro (∼15% of the total Dbx1 population) may not be representative of the situation operating in vivo because of major differences in the level of Dbx1 neuron excitation and hence ability of this network population to generate inspiratory rhythm in different states. There are sources of excitatory/neuromodulatory drive to the preBötC operating in vivo (e.g. RTN) that are absent in the slice preparations and the slices are in fact in a low excitation state (i.e., compared to the in vivo mouse, very low inspiratory discharge frequency dependent on elevated extracellular potassium for stable slow rhythm generation)*. *The conclusion that the “preBötC core is very sensitive to just a few constituent neurons (i.e. <100)” (and the authors suggestion that this may have ramifications for the loss of respiratory function in disease states) may not be valid for the in vivo state of the preBötC network, a caveat that the authors should discuss in the text. Have the authors effectively varied the state of excitation for example in the model network to gain insight into how level of excitation affects the critical mass of neurons required to sustain rhythmogenic function?*

Recognizing that measurements *in vitro* may not map directly to the preBötC *in vivo*, we acknowledge in the edited version that our quantifications apply for *in vitro* conditions only. Furthermore, we acknowledge the caveats inherent in extrapolating our *in vitro* measurements to the *in vivo* condition. Some examples:

The sentence quoted above by Reviewer #2, and three additional sentences were modified as follows, “… the preBötC core *in vitro*, is very sensitive to the loss of just a few constituent interneurons (i.e., <100).” This sensitivity to neuron loss may be accentuated in reduced slice preparations lacking excitatory and neuromodulatory drive from the rostral medulla and pons as well as peripheral chemosensory and mechanosensory vagal sensory feedback. Extrinsic sources of drive raise preBötC network excitability and enhance respiratory rhythm. An acute sensitivity to neuron loss, such as we report for slices, may not apply to the preBötC network *in vivo*, but this remains to be tested via quantitative cellular ablation experiments with physiological monitoring in intact animal models. (Discussion section)

And also in the Discussion section, “… unraveling the cellular and synaptic mechanisms of rhythmogenesis and motor output could be exploited to restore functionality in lesioned slices and, to the extent that our observations apply *in vivo*, to develop clinical therapies that bolster respiratory function in pathological conditions of animal models or human patients.”

*The results with the excitatory network model are instructive regarding perturbations of population burst frequency vs. amplitude with simulated single-cell ablation. The authors state (p. 8, para. 1) that some of their test simulation results substantiate that the model networks well represent the neonatal mouse preBötC in vitro.” In the experiments of Kam et al. 2013 cited, network bursts were generated after a substantial delay (200 - 300 ms) from the onset of neuronal activation. Does the present network model predict this temporal feature? Also the cellular burst generation and terminations mechanisms represented in the Rubin-Hayes model have not been verified experimentally, which should be noted in the Discussion. On the other hand, it could be that the type of test results used to make the claim about the model as a good representation may not be particularly sensitive to the actual burst generation and termination mechanisms incorporated*.

Almost exactly matching the experimental results of Kam *et al.* (*J Neurosci* 33: 3332-8, 2013), our network-wide evoked bursts emerged with a delay of 200-300 ms. For example, in our Figure 6 simulations, the five-cell stimulation evoked a network burst with a delay of 300.3 ms; and the six-cell stimulation evoked a network burst with a delay of 215.2 ms. 6 We respectfully point out that most of the mechanisms for burst initiation and termination in the Rubin-Hayes mathematical model (Rubin *et al. PNAS* 106: 2939-44, 2009) have in fact been experimentally tested.

Recurrent excitation: its role in initiating the inspiratory burst phase has been documented numerous times, including two canonical papers (Funk *et al. J Neurophysiol* 70: 1497-1515, 1993; Wallén-Mackenzie *et al. J Neurosci* 26: 12294-307, 2006) and three very recent papers (Kam *et al. J Neurosci* 33: 3332-8, 2013 [which used glutamate un-caging to trigger recurrent excitation]; Carroll & Ramirez. *J Neurophysiol* 109: 285-95, 2013; Carroll *et al. J Neurophysiol* 109: 296-305, 2013).

CAN current: Prof. Ramirez’ group (Univ. Wash., Seattle) was the first to document the role of *I*CAN in preBötC bursting (Thoby-Brisson & Ramirez. *J Neurophysiol* 86: 104-12, 2001), and then my lab characterized the current in some biophysical detail (Pace *et al. J Physiol* 582: 113-25, 2007; Crowder *et al. J Physiol* 582: 1047-58, 2007). Subsequently Dr. S. Mironov examined the respiratory burst-generating role of *I*CAN in a sequence of papers (Mironov *J Physiol* 586: 2277-91, 2008; Mironov & Skorova. *J Neurochem* 117: 295-308, 2011; Mironov 591: 1613-30, 2013). Finally, my group recently demonstrated the influence of *I*CAN specifically in Dbx1 preBötC neurons (Picardo *et al. J Physiol* 591: 2687-703, 2013).

Burst termination: We demonstrated the role of the electrogenic Na-K ATPase pump current (*I*pump) in burst termination in two papers (Del Negro *et al. J Physiol* 587: 1217-31, 2009; Krey *et al. Front Neural Circuits* 4: 124, 2010). Furthermore, the paper by Krey *et al.* also showed the contributions of Na^+-^dependent K+ currents (*I*Na-K) and ATP-dependent K+ currents (*I*K-ATP), which are additional activity-dependent outward currents that can contribute to burst termination as predicted by the Rubin-Hayes modeling framework (see Figure 4 of that paper).

Cellular ‘pacemaker’ activity: In the original Rubin-Hayes model, intrinsic bursting was precluded by the parameters. In a follow-up study, we explored parameter space in the model and demonstrated that the Rubin-Hayes model could show Na^+^or Ca^2+^mediated intrinsic bursting activity (Dunmyre *et al. J Comput Neurosci* 31: 305-28, 2011), which was shown experimentally several times (e.g., Thoby-Brisson & Ramirez. *J Neurophysiol* 86: 104-12, 2001; Pena *et al. Neuron* 43: 105-17, 2004; Del Negro *et al. J Neurosci* 25: 446-53, 2005).

The only aspect of the Rubin-Hayes model that has *not* yet been experimentally tested is the role of short-term synaptic depression, which also aids in burst termination in the model. My lab has been working on that problem for the last year and we are writing up the study now (although obviously we cannot cite it yet).

In the final sentence regarding the modeling, Reviewer #2 rightly questions whether the specific cellular mechanisms of the Rubin-Hayes model are important for laser ablation simulations. Presumably, if the laser ablation approach interrogates network properties, then the properties of the constituent cells may not matter. However, in our view, since the experiments are actually performed on preBötC interneurons, then the simulations should employ models that well represent preBötC neurons. Therefore, we would substitute our model for a Morris-Lecar generalized model or a stripped-down Hodgkin-Huxley generic model. For the preBötC, Rubin-Hayes is a model that we developed and thus advocate, but another contemporary model of preBötC neurons was published by Rybak’s group (Jasinski *et al. Eur J Neurosci* 37: 212-30, 2013), which builds on a model developed by Butera’s group (Toporikova & Butera. *J Comput Neurosci* 30: 515-28, 2011). We are in the process of implementing this Jasinski-Rybak model to repeat the simulations in Figure 6 and Figure 6—figure supplement 1. Unfortunately the published 7 paper by Jasinski *et al.* contains several typos and does not provide initial conditions and the code for the Jasinksi-Rybak model is not in the public domain – so it has taken my team several months just to correctly implement the single-cell version of the model and correctly reproduce the figures of their paper (to make sure the model is setup correctly). We have now begun replicating the simulations of the present study submitted to *eLife*, and we intend to publish the simulations of laser ablation using Jasinski-Rybak preBötC neuron models in a forthcoming project. We anticipate that Jasinski-Rybak will give similar results to Rubin-Hayes when integrated into network simulations and subjected to piecewise cellular ablations.

*An experimental issue to be addressed is that there is no information provided for the five slices studied in ablation experiments on the number of neurons ablated on each side of the slice. In the Materials and methods the authors state that “The domain can be bilaterally distributed like the preBötC and ventral respiratory column” and bilateral target maps are shown, but the authors should state how the targets were actually distributed in the experiments and any potential impact on the ablation results should be discussed*.

We created a new supplementary figure (Figure 2—figure supplement 2), which plots lesion tallies for preBötC-surface and control slices. It shows the total ablation tally as well as the individual tallies per side.

We now acknowledge the bilateral nature of target detection and laser ablation in the Results section..

Because Dbx1 preBötC neurons so often show commissural projections (Bouvier *et al. Nat Neurosci* 13: 1066-74, 2010; Gray *et al. J Neurosci* 30: 14883-95, 2010; Picardo *et al. J Physiol* 591: 2687-703, 2013; and Figure 8 of the present paper) we favor the null hypothesis that the connectivity of Dbx1 preBötC neurons is the same unilaterally and bilaterally. In that case, it would make no difference whether a Dbx1 neurons was ablated on side 1 or 2; the effect on network function would be the same.

Reviewer #3

[…] While the paper is clearly written, easy to follow and the figures explicit I have a few concerns:

*First the degree of novelty: the elegant method consisting in lesioning specific neuronal populations within a network and more specifically the respiratory network of the preBotzinger complex while monitoring the resulting motor output has been already presented (including all the methodological controls), (Hayes, 2012). And the important role of Dbx1-derived cells in constituting the core of the preBotzinger respiratory network has also been recently published (*[6]*;*
[20]*). Therefore the experiments dedicated to examine the effects of laser ablation of Dbx-derived respiratory neurons on the respiratory network function seem to only complement/confirm previous already published results, but do not really bring any novel results susceptible to significantly improve the current knowledge on respiratory rhythmogenesis*.

It is true that Bouvier *et al.* (*Nat Neurosci* 13: 1066-74, 2010) and Gray *et al.* (*J Neurosci* 30: 14883-95, 2010) tested the rhythmogenic role of Dbx1-derived interneurons. It is also true that our first laser ablation study (Hayes *et al. PNAS* 109: 8286-91, 2012) estimated core rhythm generator size, and established methodological controls. Nevertheless, the present study is novel and important in our view. [28] identified neurons by Ca^2+^ imaging, which cannot differentiate glutamatergic rhythm-generating neurons from GABA/glycinergic non-rhythmogenic neurons because both neuron classes oscillate in sync with the inspiratory rhythm. Moreover, Bouvier *et al.* and Gray *et al.* did not attempt to quantify the size of the Dbx1 preBötC core population or test the robustness of its functionality. Therefore, the novelty of the present study is that we test the obligatory rhythmogenic role of the Dbx1 population while quantifying its size, and in doing so we avoid counting inhibitory neurons in our measurements.

An additional element of novelty in this study is that we found that Dbx1 preBötC neurons also serve in a premotor capacity. Bouvier *et al.* and Gray *et al.* did not discover that overlapping role for Dbx1 preBötC neurons. This last point is not just a new piece of information but rather a significant paradigm shift because in the respiratory CPG a single cardinal class of interneurons serves rhythm-generating and premotor functions, whereas these functions are ordinarily attributed to separate populations.

*Second I have a problem with the use of the word “pattern”. This might be only a semantic problem but it should be made clear what the authors mean with pattern. To my point of view “pattern” means the respective and relative organization of the different phases of a rhythmic activity. For example for the respiratory behavior this could include expiration vs inspiration phases, the distinction between the different sub-phases of the inspiration activity.... But, to me, the amplitude of the hypoglossal motor output recorded from a slice preparation is not a parameter that can be associated with patterning. If we agree on this, then part of the conclusions of the paper are incorrect and should be substantially revised*.

The word ‘pattern’ was changed to ‘amplitude’, ‘magnitude, or ‘output’ as appropriate. The specific instances are marked by red text in the revised manuscript, mainly in the Abstract, Introduction, and Discussion. In all other instances the word ‘pattern’ is only used in reference to ‘central *pattern* generator’, axon projection *pattern*, or spiking *pattern*. Reviewer #1 raised a similar concern and we provide the same response.

*In the Introduction the authors claim that “rhythmogenic preBotzinger neurons are distinguished by glutamatergic neurotransmitter” and that “GABA- and glycinergic neurons in the preBotC influence sensory integration but are non-rhythmogenic because blocking inhibition does not prevent the respiratory rhythm to be generated”. This is not completely true as Morgado-Valle (2010) provided evidence that some rhythmically active glycinergic neurons of the preBotC exhibit pacemaker properties and Winter (2009) showed that glycinergic neurons are significantly integrated in the preBotC network as they constitute half of the rhythmically active neuron in this network. The authors should at least site these work here*.

Morgado-Valle *et al.* (*J Neurosci* 30: 3634-39, 2010) showed glycinergic preBötC neurons with pacemaker properties, but that does not make such neurons rhythmogenic. In fact, Morgado-Valle *et al.* used the existence of glycinergic preBötC neurons with pacemaker activity to argue that pacemaker activity *per se* is not obligatory for rhythm generation (since it is so well established that chloride-mediated synaptic inhibition, and inhibitory neurons in the preBötC, are dispensable for respiratory rhythm in slice preparations).

The paper by Winter *et al.* (*Pflügers Archiv* 458: 459-69, 2009) was cited already in the Introduction on p. 3. We also cite a large number of studies in our Introduction showing that chloride-mediated inhibition, and thus GABA- and glycinergic interneurons, are non-rhythmogenic *in vitro*. *In vivo* studies such as Büsselberg *et al.* (*Neurosci Lett* 316: 99-102, 2001) and more recently Jaczewski *et al.* (*J Neurosci* 33: 5454-65, 2013) also show that synaptic inhibition in the respiratory network is non-rhythmogenic, but rather shapes the final output pattern of respiration and influences sensorimotor integration. The Introduction now states more clearly what the role of inhibitory neurons is, “…GABA- or glycinergic neurons (37; 67), which influence sensory integration as well as coordinated patterns of muscle contraction for respiratory behaviors, but are non-rhythmogenic…”

*In the legend of*
Figure 1
*and*
Figure 4
*it is written that the red dots in panel C correspond to Dbx1- neurons and the blue ones to non-Dbx1 neurons. How does the system detect the non-Dbx1 neurons? I understood that the cell to be deleted were selected on the basis of fluorescent protein expression. By definition the non-Dbx1 neurons should not be detected because they are not tdTom positive. Thus how can they be identified by the system? Please clarify this*.

We apologize for unclear wording in the figure legends. Indeed, blue ROIs are *not* non-Dbx1 neurons. Rather, blue ROIs are rejected fluorescent targets. As we explain in the Figure 1—figure supplement 2, tdTomato-expressing ROIs are rejected as targets if they are insufficiently circular or have been previously detected in a deeper z-plane. The figure legends for 1 and 4 now read, “(C) Mask of targets showing validated Dbx1 interneuron targets (red) and regions of fluorescence that do not pass muster and were rejected as targets (blue).” We thank Reviewer #3 for pointing out this flaw in our figure legends.

*It is stated that the fluorescent neurons were preferentially selected in the dorsal part of the preBotC. Why this part more specifically? If similar experiments are performed exclusively in the ventral part of the preBotC is the ablation-induced effect on XII motor output amplitude still observed? Indeed Funk (1993), Koizumi (2013) and the present work locate the inspiratory premotoneurons in the dorsal aspect of the preBotC region, so one would expect that when ablations are performed in the ventral region of the preBotC then the consequences on motor output should be less. Did the authors ever observed a difference on results obtained depending on the ventral/dorsal position of the ablation domain? Such experiments would validate part of the results presented here and would be in complete agreement with the model used in this study*.

The patch-clamp recordings whose methods were described in the first submission were undertaken specifically to test whether Dbx1 preBötC neurons had projections to the XII nucleus, as would be expected for premotor interneurons. Reviewer #3 correctly points out that the dorsal preBötC would be the most likely position for such neurons based on [17] and [34] and 2013). Therefore, we targeted our patch recordings in the dorsal preBötC for that specific purpose, to look for Dbx1 neurons with putative premotor functionality.

If the domain for single-cell cumulative laser ablations were to be confined to the ventral preBötC, then one would quite logically predict that rhythmic frequency and stability would be perturbed, but effects on motor output amplitude would be minimized. Indeed, that would be consistent with the model we propose. We have not yet done the experiment in that way. Rather, we have taken the reverse approach and selectively laser ablated Dbx1 neurons with the domain set to the dorsal preBötC and even farther dorsally to the intermediate reticular formation (IRt). The prediction would be to degrade the amplitude of motor output without affecting frequency, and we have observed that result. These follow-up laser-ablation studies are not yet ready for publication and exceed the scope of the present manuscript in our view.